# Test-Time Adaptation by Causal Trimming

**Yingnan Liu**[1,2] **Rui Qiao**[1,3] **Mong Li Lee**[1,2] **Wynne Hsu**[1,2]
[1]School of Computing, National University of Singapore
[2]Institute of Data Science, National University of Singapore
[3]Singapore-MIT Alliance for Research and Technology
{liu.yingnan, rui.qiao}@u.nus.edu, {dcsleeml, dcshsuw}@nus.edu.sg

## Abstract

Test-time adaptation aims to improve model robustness under distribution shifts by adapting models with access to unlabeled target samples. A primary cause of performance degradation under such shifts is the model's reliance on features that lack a direct causal relationship with the prediction target. We introduce Test-time Adaptation by Causal Trimming (TACT), a method that identifies and removes non-causal components from representations for test distributions. TACT applies data augmentations that preserve causal features while varying non-causal ones. By analyzing the changes in the representations using Principal Component Analysis, TACT identifies the highest variance directions associated with non-causal features. It trims the representations by removing their projections on the identified directions, and uses the trimmed representations for the predictions. During adaptation, TACT continuously tracks and refines these directions to get a better estimate of non-causal features. We theoretically analyze the effectiveness of this approach and empirically validate TACT on real-world out-of-distribution benchmarks. TACT consistently outperforms state-of-the-art methods by a significant margin. Our code is available at `https://github.com/NancyQuris/TACT`.

## 1 Introduction

Machine learning models often exhibit significant performance degradation when evaluated on data drawn from a distribution that differs from their training data distribution [13]. To address this challenge, test-time adaptation (TTA) has emerged as a promising approach. TTA methods adapt a pretrained model to the test distribution dynamically, using the incoming test data to enhance predictive performance without requiring access to the original training data [19, 51, 56]. Despite recent advances, many existing TTA methods rely heavily on predicted labels generated by the model itself to guide the adaptation process [12, 37, 38, 51]. However, the effectiveness of these methods hinges critically on the quality of the predictions. When the model's predictions are influenced by non-causal features that do not have a direct causal relationship with the prediction target [26, 54], the predicted label may be unreliable, leading to sub-optimal adaptation outcomes [29, 47].

Unlike causal features that have stable associations with the semantic structure of the prediction task [27], non-causal features exhibit inconsistent or spurious correlations with the prediction target across training and test distributions [55]. Over-reliance on non-causal features is a key factor in model performance degradation under distribution shift. While DeYO [29] recognizes this issue, it does not explicitly mitigate reliance on non-causal features. Instead, it updates the model using predictions that leverage causal features only, relying on gradual adaptation to reinforce causal features over time. Consequently, early predictions may still be influenced by non-causal signals, requiring many adaptation steps to suppress their effects.

Given the above limitations, we propose to actively reduce non-causal features. Prior studies have shown that feature representations learned through standard training encode a mixture of causal and

non-causal features and that the causal part is often learned sufficiently well for accurate prediction [20, 27]. Motivated by this, we propose a Test-time Adaptation by Causal Trimming (TACT) framework that seeks to improve adaptation performance by isolating and removing non-causal components from the representations of samples from test distributions. Our framework aims to achieve more reliable predictions in the presence of distribution shift by reducing the model's dependence on unstable, non-causal features. To identify non-causal features in representations, we analyze how these representations change when we apply targeted perturbations to the input data. Specifically, we perform input augmentations that preserve the underlying causal contents while introducing variability in other, non-causal aspects of the input [9, 11, 18, 31, 32]. These augmentations produce multiple test-time samples that share the same causal semantics but differ in spurious or incidental attributes. By examining how the feature representations of these samples vary, we can disentangle causal and non-causal components.

We operationalize this by applying Principal Component Analysis (PCA) to the set of augmented representations and identify the direction of greatest variance. We interpret this dominant direction as being aligned with the non-causal features, under the assumption that causal content remains stable across augmentations, while non-causal attributes vary. This approach is inspired by prior work showing that high-level semantic factors are often linearly encoded in the learned representation space [1, 39, 46]. Building on the insight that linear manipulations in representation space can produce meaningful changes in semantic content [40, 50], we propose to reduce the influence of non-causal features by subtracting the projection of a test sample's representation along the identified non-causal direction. Since the prototypes used for prediction, defined as template representations corresponding to the weights for each class in the linear classifier, are influenced by non-causal features, we apply

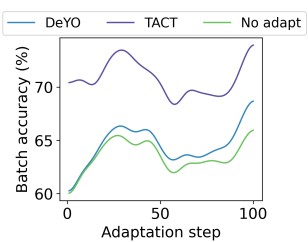

Figure 1: Batch accuracy on Camelyon17 dataset of the first 100 adaptation steps.

the same operation to them using the identified non-causal direction. During adaptation, we maintain a moving average of the updated prototypes to mitigate noise effects. Compared to DeYO, TACT can immediately produce predictions that are less affected by non-causal features, eliminating the need for iterative updates to achieve reliable results (see Figure 1). We provide a theoretical analysis to establish the conditions under which TACT can improve prediction accuracy under distribution shift. Empirically, we evaluate TACT on five real-world out-of-distribution datasets, demonstrating its effectiveness and superiority over state-of-the-art TTA methods.

## 2 Related Work

Existing TTA methods can be broadly categorized into backpropagation-free and backpropagation-based methods. Backpropagation-free methods modify model outputs or intermediate representations without gradient-based optimization. These include modifiable prompts [36], re-normalized representations [44], updated prototypes [19, 57], and maximum likelihood estimation [4]. Backpropagation-based methods update the model with the gradient of objective functions such as entropy minimization [12, 37, 38, 51] and self-training with pseudo-labels [17, 25, 47, 52]. Entropy Minimization encourages more confident predictions by reducing the entropy of model predictions during adaptation. Self-training employs cross entropy [5, 17, 30, 47] and knowledge distillation [25, 52, 53] using model predictions as pseudo-labels. Regularization measures such as information maximization [30], representation statistics alignment [23, 58], and consistency regularization [35, 56] for invariant prediction under augmentations have been proposed to regularize the adaptation.

A key challenge in test-time adaptation is obtaining reliable pseudo-labels to guide model updates. One line of work assumes that correct predictions tend to exhibit low entropy, and update the model using only high-confidence samples with low-entropy predictions [19, 37, 38, 57]. However, DeYO [29] shows that spurious correlations can also result in low entropy predictions and proposes a causal intervention technique to identify predictions that are more likely based on causal features, using them selectively for model updates. Another line of work refines pseudo-labels by incorporating updated prototype and neighborhood information [5, 17, 21, 47, 53]. AdaContrast [5] uses soft voting among nearest neighbors. TSD [53] relies on updated prototypes and spatial local clustering. TAST [21] employs neighbourhood information in self-training. PROGRAM [47] considers both prototype

and neighbour-based pseudo-labels to enhance label quality. PASLE [17] progressively refines the pseudo-labels of uncertain predictions using updated prototypes.

All the above methods, except for DeYO, do not consider the effect of non-causal features on model prediction. Although DeYO finds that non-causal features would make entropy an unreliable metric to reflect prediction correctness, it does not adjust model predictions. TACT adjusts model predictions by reducing non-causal features, and our adjusted prediction can be used as a more reliable pseudo-label.

# 3 Preliminaries

We consider the problem of adapting a well-trained model to test-time distributions that differ from the training distribution. Our goal is to improve the model's performance on these shifted distributions with unlabeled test samples. Following prior work [18, 48], we model the distribution shift using a structural causal model that captures the underlying data-generating process, as illustrated in Figure 2.

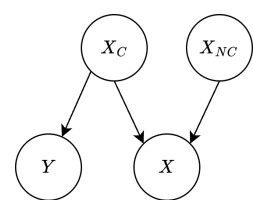

Figure 2: Structural causal model of the data-generating process.

We model the observed sample $X$ and its label $Y$ as being generated from causal factors $X_C$ and non-causal factors $X_{NC}$. Only $X_C$ is causally related to $Y$, while $X$ is related to both $X_C$ and $X_{NC}$. The correlation between $X_C$ and $Y$ is stable, i.e., the conditional distribution $P(Y|X_C)$ remains unchanged at test time. We also assume that the distribution of causal factors $P(X_C)$ remains invariant across the training and test datasets, whereas distribution shifts arise from changes in $P(X_{NC})$. The model would have stable performance across distributions if the prediction is based solely on features related to the causal factors $X_C$. In contrast, reliance on non-causal factors $X_{NC}$ can lead to unreliable performance under distribution shifts.

We consider a $c$-class classification task, where the model $f := h \circ g$ used for adaptation is composed of a feature extractor $g$ and a classifier $h$. The feature extractor $g$ maps an input sample to a $d$-dimensional vector $z \in \mathbb{R}^d$ as the representation. The classifier $h$ maintains a set of class prototypes $\{q_1, \ldots, q_c\} \in \mathbb{R}^d$, where each prototype $q_i$ serves as a template representation for class $i$. Predictions are made by computing the similarity between the input representation $z$ and each class prototype using the dot product $z \cdot q_i$, referred to as the logit of class $i$. A softmax function is then applied to the logits to obtain the probability distribution over the classes.

# 4 Proposed Method

The works in [20, 27] show that models are often capable of learning causal features, even when their predictions are predominantly driven by non-causal features with spurious correlations. However, the predictive influence of these causal features is frequently obscured or suppressed due to the heavily weighted non-causal components in the learned representations. Based on this observation, we propose TACT, a method that identifies non-causal features and reduces their influence by causally trimming the learned representations. We hypothesize that non-causal features are embedded in representations along a specific direction. Such direction is of the maximum variance when the non-causal features change. To suppress their influence, we subtract the projection of both the input representation and class prototypes onto this identified direction. This operation attenuates the non-causal information present in both elements. Since class prototypes serve as canonical representations for each class, and the non-causal direction estimated from a single test sample may be noisy, we maintain a moving average of the updated prototypes throughout test-time adaptation. At inference, predictions are made by measuring the similarity between the adapted representation and the moving average of the updated prototypes, thereby reducing influence of non-causal features.

## 4.1 Non-Causal Feature Identification

Given a sample $x$ at test time, if we have access to additional samples generated with the same causal factors but different non-causal factors, we can compare their representations to infer the influence of non-causal factors. Changes in the representations across these samples can be attributed to variations

in these non-causal factors. By systematically analyzing these representational differences, we can isolate and identify the components of the representation that correspond to non-causal features.

To simulate variations in the data-generating process, we apply data augmentation to target non-causal features [9, 11, 18, 31, 32]. For a test sample $x$, we generate $n$ augmented samples $\{\tilde{x}_i\}_{i=1}^n$ that preserve the causal feature while varying the non-causal factors. We collect the representations of these samples in a matrix $\mathbf{Z} = [z, \tilde{z}_1, ..., \tilde{z}_n]^\top$, where $z$ is the representation of the original sample and $\tilde{z}_i$ are those of the augmented samples.

We interpret non-causal features as corresponding to specific, disentangled directions in the representation space, consistent with prior work that indicates high-level semantic concepts are linearly encoded as vector directions in learned representations [1, 39, 46]. For instance, the vector difference between "woman" and "man" would resemble that between "queen" and "king" [33], both aligning to the direction representing gender. Along this direction, specific instances of the gender concept, such as "male" and "female", take different magnitudes.

Given representations of samples that differ only in their non-causal factors, the direction along which the representations change the most is expected to capture the non-causal features. This dominant direction can be identified via Principal Component Analysis (PCA) which analyzes the covariance matrix of the representations to extract the principal components. Principal components are vectors along which the representations' projections exhibit maximum variances. We first compute the mean of the representations as: $\bar{z} = \frac{1}{n+1}z + \frac{n}{n+1}\sum_{i=1}^n \tilde{z}_i$. Using this mean, we construct a matrix $\bar{\mathbf{Z}} = [\bar{z}, \bar{z}, .., \bar{z}]^\top$ that has the same size as the representation matrix $\mathbf{Z}$. The covariance matrix of the representations is then given by $\mathbf{\Sigma_Z} = (\mathbf{Z} - \bar{\mathbf{Z}})^\top (\mathbf{Z} - \bar{\mathbf{Z}})$. The eigenvectors of $\mathbf{\Sigma_Z}$ correspond to the principal components, and their eigenvalues quantify the variance along these components [22]. Since $\mathbf{\Sigma_Z}$ is a real symmetric matrix, its eigenvectors form an orthonormal basis in $\mathbb{R}^d$ [16]. Using spectral decomposition, we express the covariance matrix as: $\mathbf{\Sigma_Z} = \mathbf{Q}\mathbf{\Lambda}\mathbf{Q}^\top$, where $\mathbf{Q} = [e_1, e_2, ..., e_d]$ is an orthogonal matrix whose columns are the orthonormal eigenvectors, and $\mathbf{\Lambda}$ is a diagonal matrix containing the eigenvalues of $\mathbf{\Sigma_Z}$. Here, $e_i$ denotes the direction along which the variance of the projected representations is the $i^{th}$ largest.

## 4.2 Causal Trimming to Reduce Non-Causal Feature

Prior work has demonstrated that applying linear transformations to representations can manipulate the semantics they encode [40, 50]. Since the principal components $\{e_i\}_{i=1}^d$ form an orthonormal basis in $\mathbb{R}^d$, any representation $z$ can be expressed as a linear combination of these components. To reduce the influence of non-causal features, we propose to trim the representation by removing its components along the top-$m$ principal components:

$$\hat{z} = z - \sum_{i=1}^m (z \cdot e_i)e_i \tag{1}$$

Since each $e_i$ is a vector of unit length, the term $(z \cdot e_i)e_i$ is the projection of $z$ onto $e_i$ whose magnitude is given by the dot product $(z \cdot e_i)$. By subtracting these terms, we obtain an updated representation $\hat{z}$ which is composed of components only formed by $\{e_i\}_{i=m+1}^d$. If causal features are invariant under data augmentations and their corresponding semantic directions are orthogonal to those of the removed directions, causal information present in $z$ is preserved in the trimmed representation $\hat{z}$.

## 4.3 Model Adaptation

In a prototype-based classifier, each class prototype $q_j$ serves as a template representation learned by the classifier $h$, summarizing the representations of samples belonging to class $j$. However, if the learned representations encode non-causal features, the prototypes will be influenced by these features. To mitigate this issue, we apply the same causal trimming to the class prototypes. Specifically, let $q_j$ be the prototype of class $j$. Given the top-$m$ principal components $\{e_i\}_{i=1}^m$ that are used to trim the test sample representation $z$, we obtain the trimmed prototype $\hat{q}_j$ for each class $j \in \{1, 2, \ldots, c\}$ as:

$$\hat{q}_j = q_j - \sum_{i=1}^m (q_j \cdot e_i)e_i \tag{2}$$

Since the identified non-causal directions may vary across samples due to noise or context-specific factors, we compute a batch-wise average of trimmed prototypes to obtain a more stable estimate. To track the estimate across batches during adaptation, we maintain a moving average of the trimmed prototypes during test-time adaptation. Suppose we obtain trimmed prototypes $\hat{q}_j^{(i)}$ for each class $j$ at batch $i$, the moving-average $\bar{\hat{q}}_j$ is updated by $\bar{\hat{q}}_j = \frac{i-1}{i}\bar{\hat{q}}_j + \frac{1}{i}\hat{q}_j^{(i)}$. This moving average serves as a more robust estimate of the causally refined prototypes, effectively smoothing out sample-specific variance. The prediction made by the average of the trimmed prototypes is the same as that of an ensemble over the logits produced by individual trimmed prototypes, resulting in more stable predictions. At test time, for a given input sample $x$, we compute the causally trimmed representation $\hat{z}$ and compare it to the moving-averaged trimmed prototypes $\bar{\hat{q}}_j$. The logit for class $j$ is given by the dot product $\hat{z} \cdot \bar{\hat{q}}_j$, and the final predicted label $y$ is given by:

$$y = \arg\max_j \frac{\exp\left(\hat{z} \cdot \bar{\hat{q}}_j\right)}{\sum\limits_{i=1}^{c} \exp\left(\hat{z} \cdot \bar{\hat{q}}_i\right)} \tag{3}$$

## 5 Theoretical Analysis

We present the conditions under which TACT would correct a wrong prediction and maintain a correct prediction. We consider binary classification $Y \in \{+1, -1\}$. The two prototypes learned by the binary classifier $h$ are represented as $\{q_{+1}, q_{-1}\}$. We drop the bias term for simplicity. Meanwhile, we assume the existence of causal prototypes $\{p_{+1}, p_{-1}\}$, which always make correct predictions on the learned representations and do not leverage non-causal features. To simplify the analysis, we consider the decision boundary vectors $\Delta q = q_{+1} - q_{-1}$ and $\Delta p = p_{+1} - p_{-1}$. We analyze the representation $z$ of an instance with label $y$. Given the principal components $\{e_i\}_{i=1}^{d}$ computed from $z$ and its augmented variants, we write $z$ as $\sum_{i=1}^{d} \alpha_i e_i$, where $\alpha_i$ is the magnitude of $z$'s projection on $e_i$. Similarly, we define the learned decision boundary $\Delta q$'s projection magnitude as $\{\gamma_i\}_{i=1}^{d}$. We write the projection magnitude of causal decision boundary $\Delta p$ as $\{\eta_i \gamma_i\}_{i=1}^{d}$, to view $\Delta p$ as a transformation from $\Delta q$ by a projection magnitude $\eta_i$ on the direction of each principal component.

We can obtain $\hat{z}$ by trimming the top-$m$ principal components (PCs) for $z$. Proposition 1 shows the conditions under which TACT can correct a wrong prediction.

**Proposition 1.** *For any $z$ that is misclassified by the learned decision boundary $\Delta q$, the misclassification can be corrected by using the representation obtained after removing the top-$m$ principal components, if both of the following two conditions are satisfied:*

$$y \sum_{i=1}^{m} \alpha_i \gamma_i < 0 \quad and \quad y \sum_{i=m+1}^{d} \alpha_i \gamma_i > 0 \tag{4}$$

$$\left| \sum_{i=1}^{m} \alpha_i \gamma_i \right| > \left| \sum_{i=m+1}^{d} \alpha_i \gamma_i \right| \tag{5}$$

Appendix A.1 provides the formal proof. Equation (4) captures the case in which a prediction based solely on the top-$m$ PCs leads to an incorrect outcome, whereas a prediction based on the remaining PCs yields the correct result. Equation (5) requires the absolute value of the prediction score derived from the top-$m$ PCs must be greater than that from the remaining PCs. Together, these conditions in Proposition 1 suggests that a wrong prediction can be corrected by TACT when the top-$m$ PCs are solely responsible for the wrong prediction, and the prediction made by the top-$m$ PCs weighs more than the prediction made by the remaining PCs.

In Proposition 2, we establish the conditions under which the trimmed representation $\hat{z}$ retains sufficient causal information to preserve the correct prediction by the causal decision boundary $\Delta p$.

**Proposition 2** (Causal Preservation). *For any original representation $z$, the trimmed representation $\hat{z}$ preserves the correct prediction under the causal decision boundary $\Delta p$ if any one of the following*

*conditions holds:*

$$\begin{cases} y \sum_{i=1}^{m} \eta_i \alpha_i \gamma_i = 0 \\ y \sum_{i=1}^{m} \eta_i \alpha_i \gamma_i < 0 \\ 0 < y \sum_{i=1}^{m} \eta_i \alpha_i \gamma_i < y \sum_{i=1}^{d} \eta_i \alpha_i \gamma_i \end{cases} \tag{6}$$

The proof is provided in Appendix A.2. Equation (6) characterizes three cases: (a) the top-$m$ PCs have no contribution to the causal prediction; (b) the top-$m$ PCs has a negative influence on the causal prediction and thus their removal is beneficial; (c) the top-$m$ PCs has a positive contribution, but the representation forms by all PCs contribute even more strongly. When the top-$m$ PCs have no contribution to the causal predictions, they are considered non-causal features. In other words, the removed component $z - \hat{z}$ does not contain causal information. When the top-$m$ PCs contain causal information, $m$ should be selected such that the top-$m$ PCs contribute less to the prediction compared to all the PCs, ensuring that the trimmed representation $\hat{z}$ remains causally informative. In other words, sufficient causal features need to be preserved after causal trimming.

Finally, in Proposition 3, we identify the conditions under which causal trimming would have no negative impact on the prediction of samples that are already correctly classified.

**Proposition 3.** *Suppose $z$ is correctly classified by the learned decision boundary $\Delta q$. The trimmed representation $\hat{z}$ obtained via TACT will still be classified correctly if either of the conditions holds:*

1. $y(z - \hat{z})\Delta q \leq 0$, *or*

2. $y(z - \hat{z})\Delta q > 0$, *and Equation* (7) *holds, assuming $\hat{z}$ already satisfies the Causal Preservation condition (Proposition 2).*

$$\text{sign}\left( \sum_{i=m+1}^{d} \eta_i \alpha_i \gamma_i \right) = \text{sign}\left( \sum_{i=m+1}^{d} \alpha_i \gamma_i \right) \tag{7}$$

The proof can be found in Appendix A.3. Equation (7) indicates that when classification relies only on the representations formed by the remaining PCs, the learned decision boundary makes the same prediction as the causal decision boundary. Proposition 3 also shows that if a correct prediction is made by the learned decision boundary, TACT will preserve this correctness as long as the removed part $z - \hat{z}$ contributes negatively or does not contribute to the prediction. On the other hand, when the trimmed representation $\hat{z}$ contains sufficient causal information as established in Proposition 2, the learned decision boundary is required to align directionally with the causal decision boundary defined by the remaining PCs.

## 6 Performance Study

We study the test-time adaptation performance under real-world distribution shifts, using datasets from multiple modalities, including image, audio, and text. Compared to prior works that primarily benchmark on image data, our comprehensive experiments offer broader insights into the generalizability of TACT and other TTA methods.

**Datasets.** We summarize the datasets used in our experiments below:

- Birdcalls [15, 24, 34], curated by [9], is an audio classification dataset to identify bird species from clips recorded in diverse environments. Each clip is converted into a Mel spectrogram for classification. Distribution shifts stem from variations in microphone gain settings, habitat acoustics (e.g. other animal sounds), and bird population. The test set includes 724 audio clips.
- Camelyon17 [2], sourced from from the Wilds benchmark [28], is a medical imaging dataset for binary classification of tumor versus normal tissue images. The distribution shift arises from variations in slide staining protocols, patient demographics, and scanner equipment. The test set consists of 85,054 images.
- CivilComments [3], from the Wilds benchmark [28], is a natural language dataset comprising user-submitted text comments. The task is to classify whether a comment is toxic or non-toxic.

The toxicity is spuriously associated with the mention of certain demographics in the training data. The test set contains 133,782 comments.

- ImageNet-R [14] contains 30,000 images of objects from 200 ImageNet [42] classes. The images consist of various renditions, resulting in visual domain shifts from the original dataset.
- ImageNet-V2 [41] is collected years after the original ImageNet using the same methodology, and includes 10,000 images across 1,000 original classes. It represents a natural temporal shift.

**Non-causal feature identification for TACT.** We applied the following data augmentations to identify non-causal features in each dataset: For Birdcalls, we follow [9] that investigates augmentations that randomize features independent of labels but dependent on distributions. Here, random color jitter is applied to the Mel spectrograms to simulate changes in microphone gain settings. For Camelyon17, we use stain color jitter [49] as suggested in [9] to mimics variations in histopathological slide staining. For CivilComments, we randomly prepend or append short demographic-referencing sentences to the original text. The full list of sentences is provided in Appendix B. For ImageNet-R and ImageNet-V2, where the sources of distribution shift are unknown, we experiment with general-purpose image augmentations. Specifically, we apply AutoAugment [6] with ImageNet policy and RandomAugment [7]. Both methods apply a series of transformations to the images. A detailed discussion on augmentation design and selection in practice is presented in Appendix C.

**Baselines.** Since TACT is a backpropagation-free approach, we compare TACT with the following state-of-the-art (SOTA) TTA backpropagation-free algorithms:

- T3A [19] adapts the classifier by updating class prototypes using confident test-time representations.
- LAME [4] adjusts model output probabilities via Laplacian-adjusted maximum likelihood estimation.
- FOA [36] introduces an adaptable prompt at model input to match the representation statistics of test and train data.

We also implement a variant called TACT-adapt, where predictions from TACT are used to guide gradient-based model updates with cross entropy loss $\mathcal{L}_{CE}$. We employ the information maximization loss $\mathcal{L}_{IM}$ proposed in SHOT [30] as regularization. We optimize the model using the objective: $\mathcal{L} = \mathcal{L}_{CE}(\hat{y}, y_{\text{TACT}}) + \lambda \mathcal{L}_{IM}(\hat{y})$. $\hat{y}$ is the model's prediction, and $y_{\text{TACT}}$ is TACT's prediction. $\lambda$ is the hyperparameter balancing the two terms.

We compare TACT-adapt with the following SOTA backpropagation-based methods:

- SHOT [30] adapts the feature extractor using information maximization and cross entropy loss on confident prediction.
- Tent [51] performs entropy minimization to update the affine parameters of normalization layers at test time.
- SAR [38] builds upon Tent by incorporating sharpness-aware minimization and model reset to mitigate overfitting to noisy samples.
- DeYO [29] identify confident samples that leverage causal features only by image augmentations that destroy shapes and using confidence-reweighted entropy minimization to update the affine parameters.
- TAST [21] adapts a trainable module on top of the trained feature extractor via self-training with nearest neighbor information.
- TSD [53] enhances feature representations through self-distillation and local clustering, ensuring alignment and uniformity while filtering noisy labels.
- PASLE [17] refines uncertain pseudo-labels progressively using selective label enhancement with candidate label sets and classifier-consistent loss.

**Model architecture.** We study TACT on transformer-based architectures, which are increasingly used in practice but remain relatively underexplored in TTA. Specifically, we use ViT-B/32 [8] as the backbone for Birdcalls, Camelyon17, ImageNet-R, and ImageNet-V2, and DistilBERT [43] for CivilComments. Appendix D.1 provides more details on model studied.

**Hyperparameters and model selection.** We use a test batch size of 64 [29, 36]. There are two hyperparameters in TACT, the number of augmentation $n$ and the number of removed principal components $m$. We search $n \in \{2^1, 2^2, \ldots, 2^8\}$, $m \in [1, 16]$ and $m$ is an integer. For TACT-adapt, we search $\lambda \in \{1, 5\} \times \{0.1, 1, 10, 100\}$. The rest hyperparameters follow the search space

Table 1: Test-time adaptation performance (%). We group the methods into backpropagation-free (BP-free) and backpropagation-based (BP-based). The best performance of each dataset is in bold.

| | Method | Birdcalls | Camelyon17 | CivilComments | ImageNet-R | ImageNet-V2 |
|---|---|---|---|---|---|---|
| | No TTA | 22.74 | 62.31 | 55.38 | 41.83 | 62.97 |
| BP-free | T3A | 26.16±1.33 | 69.96±1.98 | 56.43±0.00 | 41.78±0.12 | 62.93±0.02 |
| | LAME | 23.66±1.01 | 62.38±0.03 | 56.24±0.10 | 41.77±0.01 | 63.00±0.02 |
| | FOA | 26.95±1.81 | 58.36±0.77 | - | 41.46±0.16 | 62.76±0.08 |
| | **TACT** | 31.14±1.69 | 70.17±0.05 | 71.80±0.35 | 43.59±0.02 | 63.33±0.10 |
| BP-based | SHOT | 26.82±5.14 | 80.28±5.61 | 13.93±0.97 | 48.79±0.08 | 63.32±0.09 |
| | Tent | 23.16±0.42 | 62.29±0.01 | 55.38±0.00 | 42.08±0.05 | 63.09±0.03 |
| | SAR | 23.16±0.42 | 62.30±0.00 | 55.38±0.00 | 42.58±0.11 | 62.97±0.01 |
| | DeYO | 23.29±0.39 | 69.64±1.47 | - | 46.87±0.08 | 62.96±0.01 |
| | TAST | 26.08±1.11 | 83.01±1.42 | 56.56±0.20 | 41.09±0.08 | 62.84±0.07 |
| | TSD | 27.33±1.75 | 67.33±4.74 | 55.38±0.00 | 41.76±0.01 | 62.98±0.01 |
| | PASLE | 27.35±1.79 | 60.66±0.04 | 55.77±0.15 | 46.08±0.09 | 63.15±0.04 |
| | **TACT-adapt** | **31.25±3.59** | **83.70±1.10** | **71.98±0.19** | **48.81±0.05** | **63.44±0.07** |

of SHOT. For all baseline methods, we perform hyperparameter tuning within the search spaces specified in their respective papers. The detailed configurations and search procedures are provided in Appendix D.2. Following the protocol recommended in [59], we employ oracle selection to choose the best-performing hyperparameters, ensuring a fair and consistent evaluation across all methods.

## 6.1 Test-time Adaptation Performance

Following the evaluation protocol of each dataset, we use macro F1 for Birdcalls, accuracy for Camelyon17, ImageNet-R and ImageNet-V2, and worst-group accuracy for CivilComment, whose data are grouped by demographic attributes and toxicity. Due to the high variability observed in Birdcalls, each experiment is repeated ten times, whereas experiments on the remaining datasets are conducted three times. The mean and standard deviation are summarized in Table 1.

We see that TACT consistently outperforms existing backpropagation-free methods on all the datasets, with substantial gains of 4% on Birdcalls, 15% on CivilComments, and 1.7% on ImageNet-R. Further, TACT-adapt achieves the best overall performance across all datasets, outperforming both backpropagation-free and backpropagation-based baselines. These results suggest that non-causal features are a major source of performance degradation under distribution shift, and that removing them improves predictive reliability. It also confirms the value of TACT not only as a standalone method but also as a reliable supervisory signal for test-time learning.

We note that TACT performs well when causal features are approximately invariant under augmentation. For ImageNet-R and ImageNet-V2, AutoAugment [6] and RandomAugment [7] maintain the key causal features, which are object structure and shape [10, 29]. Other causal features that could be helpful in inferring objects, such as color when inferring strawberries, are altered. In addition, the models we perform adaptation on do not have their representation space explicitly constrained such that causal and non-causal features are linearly encoded, disentangled, or orthogonal. Yet, the approximate separation of causal and non-causal features by PCA yields consistent performance gains, suggesting the robustness of TACT.

## 6.2 Visualization of Predictions after Causal Trimming

To gain insight into the predictions made after causal trimming, we employ GradCAM [45] to visualize the focus of the original predictions and those made by TACT on samples from ImageNet-R. GradCAM identifies which parts of an input image contribute most to a prediction by computing the gradients of the predicted class score with respect to the embeddings of the image patches. The resulting heatmaps are overlaid on the input images, where brighter regions indicate higher importance for the prediction.

The visualization results are presented in Figure 3. Compared to the original predictions, TACT places less emphasis on non-causal information, such as background elements. For instance, in the

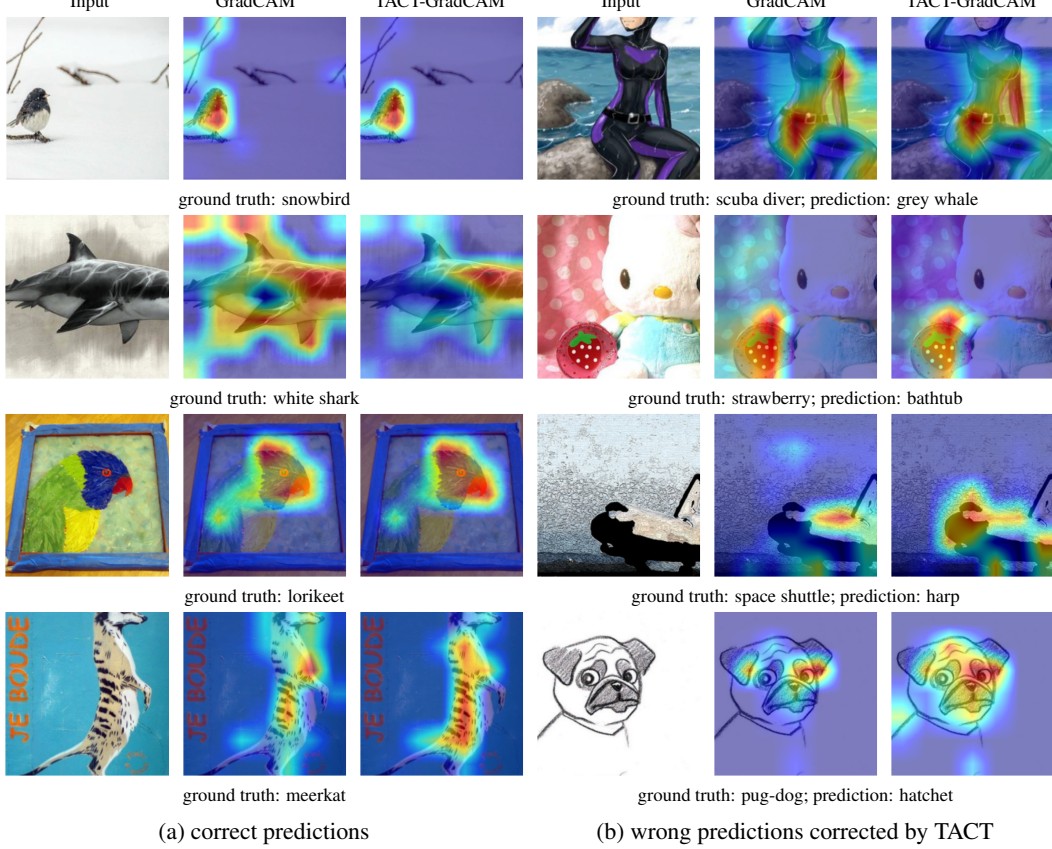

| Input | GradCAM | TACT-GradCAM | Input | GradCAM | TACT-GradCAM |

ground truth: snowbird

ground truth: scuba diver; prediction: grey whale

ground truth: white shark

ground truth: strawberry; prediction: bathtub

ground truth: lorikeet

ground truth: space shuttle; prediction: harp

ground truth: meerkat

ground truth: pug-dog; prediction: hatchet

(a) correct predictions          (b) wrong predictions corrected by TACT

Figure 3: GradCAM visualizations of the original predictions and TACT's predictions.

snowbird sample, TACT disregards irrelevant features like the surrounding branches. Similarly, in the white shark example, TACT restricts the focus to the object itself, unlike the original prediction that diffuses significant attention across the background. The sea background in the scuba diver example, and the dot texture in the background of the strawberry example, are likely to be spuriously correlated with certain prediction classes. These features are de-emphasized by TACT, contributing to a more accurate prediction.

Furthermore, TACT enhances attention on core causal features, leading to a sharper focus on an object's defining characteristics. This is clearly demonstrated in the lorikeet example, where the beak becomes the key focus, and the meerkat example, where attention is concentrated on the banded pattern and body. Moreover, in cases where the original prediction neglects causal features, as shown in the space shuttle and pug-dog example, TACT can redirect the emphasis to the actual salient features, such as the nose cone of the space shuttle and the face of the pug-dog, resulting in improved prediction performance.

## 6.3 Effect of Hyperparameters

The performance of TACT depends on two key hyperparameters: the number of augmentations $n$ and the number of removed principal components $m$. These parameters govern the accuracy of non-causal direction estimation and the extent of causal trimming, respectively. Figure 4 shows the performance under different numbers of augmentations and removed principal components for the Camelyon17, CivilComments and ImageNet-R datasets.

Since representations from augmented samples are used to compute the covariance matrix from which the directions of maximum variances are identified, a larger number of augmentations $n$ generally leads to more stable and accurate identification of non-causal directions. Empirically, we find that values of $n \in \{128, 256, 512\}$ provides sufficient performance, while small values of $n$ often fail to

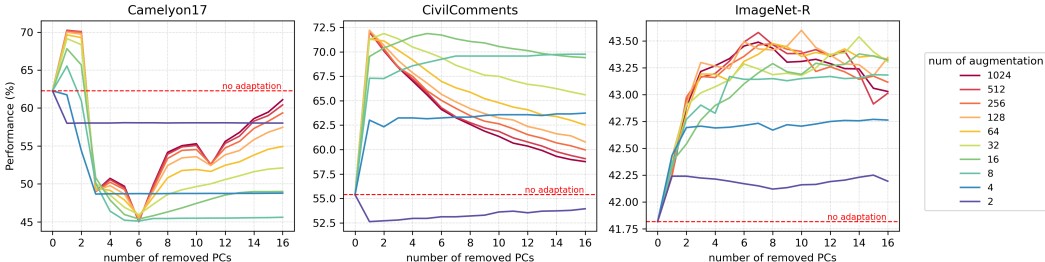

Figure 4: Performance across number of augmentation and number of removed principal components.

adequately capture the variance needed for accurate principal component estimation. The number of removed principal components $m$, should be carefully selected to ensure effective reduction of non-causal features while retaining sufficient causal features as suggested by our theoretical analysis. In practice, removing the top principal component, which typically captures the dominant non-causal variation, often suffices. However, for datasets with more complex or layered distribution shifts, such as ImageNet-R, removing more principal components would further boost the performance.

## 6.4 Ablation Study

We conduct two sets of ablation experiments. In the first experiment, we isolate the impact of representation trimming, without prototype averaging. In the second experiment, we assess whether prototype averaging alone can sufficiently filter out non-causal features.

Table 2 shows the results. We observe that trimming the representation $z$ yields better performance than no adaptation, confirming that removing components aligned with non-causal directions in representations is beneficial. While using only the averaged trimmed prototypes $\hat{q}$ also improves performance over no adaptation, the gains are generally less significant than when trimmed representations are employed. This suggests that relying solely on the averaged trimmed prototypes is insufficient for effectively reducing non-causal features. The best performance is achieved when both the trimmed representation and the averaged trimmed prototypes are used in conjunction, indicating that mitigating non-causal features in both representations and prototypes is crucial.

Table 2: Results of ablation study.

| trim $z$ | trim $q$ | average $\hat{q}$ | Birdcalls | Camelyon17 | CivilComments | ImageNet-R | ImageNet-V2 |
|:---:|:---:|:---:|:---:|:---:|:---:|:---:|:---:|
| | No TTA | | 22.74 | 62.31 | 55.38 | 41.83 | 62.97 |
| ✓ | | | 25.91±1.67 | 69.43±0.01 | 67.84±0.37 | 43.21±0.03 | 63.24±0.10 |
| | ✓ | ✓ | 27.36±0.23 | 64.74±0.05 | 62.41±0.08 | 42.24±0.00 | 63.03±0.01 |
| ✓ | ✓ | ✓ | **31.14±1.69** | **70.17±0.05** | **71.80±0.35** | **43.59±0.02** | **63.33±0.10** |

## 7 Conclusion and Future Work

We present TACT, a test-time adaptation method that reduces model reliance on non-causal features for test representations. TACT identifies non-causal components in the representation space by analyzing samples with identical causal features but varying non-causal features. The directions of maximum variance among the representations are treated as the non-causal directions. To adapt the model, we subtract the projection of the representation and class prototypes onto this non-causal direction. We keep track of the identified directions and utilize the average of the trimmed class prototypes for improved prediction. We analyze the theoretical conditions for TACT to enhance predictive performance. Extensive experiments on five real-world out-of-distribution datasets demonstrate the effectiveness and generalizability of our approach. While TACT demonstrates strong performance, it requires prior knowledge of the data to select augmentations that vary non-causal features without altering causal ones. Future work should explore identifying non-causal features when such knowledge is unavailable, and better methods to find non-causal features beyond PCA's orthogonality constraint.

## Acknowledgement

We thank Dr Fusheng Liu for the helpful discussions. We appreciate the anonymous reviewers and AC for the constructive and valuable feedback.

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

# A  Details of Theoretical Analysis

## A.1  Conditions for TACT to correct a wrong prediction

We first restate Proposition 1 as follows:

**Proposition 1.** *For any $z$ that is misclassified by the learned decision boundary $\Delta q$, the misclassification can be corrected by using the representation obtained after removing the top-$m$ principal components, if both of the following two conditions are satisfied:*

$$y \sum_{i=1}^{m} \alpha_i \gamma_i < 0 \quad and \quad y \sum_{i=m+1}^{d} \alpha_i \gamma_i > 0 \tag{4}$$

$$\left| \sum_{i=1}^{m} \alpha_i \gamma_i \right| > \left| \sum_{i=m+1}^{d} \alpha_i \gamma_i \right| \tag{5}$$

*Proof.* As the learned decision boundary $\Delta q$ cannot classify $z$ correctly, we have:

$$yz \cdot \Delta q < 0$$

$$y \sum_{i=1}^{d} \alpha_i e_i \cdot \sum_{i=1}^{d} \gamma_i e_i < 0$$

$$y \sum_{i=1}^{d} \alpha_i \gamma_i (e_i \cdot e_i) < 0$$

$$y \sum_{i=1}^{d} \alpha_i \gamma_i < 0$$

$$y \sum_{i=1}^{m} \alpha_i \gamma_i + y \sum_{i=m+1}^{d} \alpha_i \gamma_i < 0 \tag{8}$$

TACT updates $z$ to $\hat{z}$ and $q$ to $\hat{q}$ via causal trimming, and the resulting prediction is correct if and only if $y\hat{z} \cdot \Delta\hat{q} > 0$, which leads to:

$$y\hat{z} \cdot \Delta\hat{q} > 0$$

$$y \sum_{i=m+1}^{d} \alpha_i e_i \cdot \sum_{i=m+1}^{d} \gamma_i e_i > 0$$

$$y \sum_{i=m+1}^{d} \alpha_i \gamma_i (e_i \cdot e_i) > 0$$

$$y \sum_{i=m+1}^{d} \alpha_i \gamma_i > 0 \tag{9}$$

By combining Equation (8) and (9), we can derive:

$$y \sum_{i=1}^{m} \alpha_i \gamma_i < -y \sum_{i=m+1}^{d} \alpha_i \gamma_i < 0 \tag{10}$$

In addition:

$$\left| \sum_{i=1}^{m} \alpha_i \gamma_i \right| > \left| \sum_{i=m+1}^{d} \alpha_i \gamma_i \right| \tag{11}$$

$\square$

## A.2 Conditions for trimmed representations to preserve causal features

**Proposition 2** (Causal Preservation). *For any original representation $z$, the trimmed representation $\hat{z}$ preserves the correct prediction under the causal decision boundary $\Delta p$ if any one of the following conditions holds:*

$$
\begin{cases}
y \sum\limits_{i=1}^{m} \eta_i \alpha_i \gamma_i = 0 \\
y \sum\limits_{i=1}^{m} \eta_i \alpha_i \gamma_i < 0 \\
0 < y \sum\limits_{i=1}^{m} \eta_i \alpha_i \gamma_i < y \sum\limits_{i=1}^{d} \eta_i \alpha_i \gamma_i
\end{cases}
\tag{6}
$$

Equation (6) characterizes three cases: (a) the top-$m$ PCs have no contribution to the causal prediction; (b) the top-$m$ PCs has a negative influence on the causal prediction and thus their removal is beneficial; (c) the top-$m$ PCs has a positive contribution, but the representation forms by all PCs contribute even more strongly. When the top-$m$ PCs have no contribution to the causal predictions, they are considered non-causal features. In other words, the removed component $z - \hat{z}$ does not contain causal information. When the top-$m$ PCs contain causal information, $m$ should be selected such that the causal information in the top-$m$ PCs contributes less to the prediction compared to all the PCs, ensuring that the trimmed representation $\hat{z}$ remains causally informative.

The proof provided here corresponds to this corrected version.

*Proof.* As the causal decision boundary $\Delta p$ can classify $z$ correctly, we have:

$$
yz \cdot \Delta p > 0
$$

$$
y \sum_{i=1}^{d} \alpha_i e_i \cdot \sum_{i=1}^{d} \eta_i \gamma_i e_i > 0
$$

$$
y \sum_{i=1}^{d} \eta_i \alpha_i \gamma_i (e_i \cdot e_i) > 0
$$

$$
y \sum_{i=1}^{d} \eta_i \alpha_i \gamma_i > 0
$$

$$
y \sum_{i=1}^{m} \eta_i \alpha_i \gamma_i + y \sum_{i=m+1}^{d} \eta_i \alpha_i \gamma_i > 0
\tag{12}
$$

By rearranging Equation (12), we can derive:

$$
y \sum_{i=m+1}^{d} \eta_i \alpha_i \gamma_i > -y \sum_{i=1}^{m} \eta_i \alpha_i \gamma_i
\tag{13}
$$

Using causal decision boundary to predict $\hat{z}$, the prediction is correct if and only if $y\hat{z} \cdot \Delta p > 0$, which leads to:

$$
y\hat{z} \cdot \Delta p > 0
$$

$$
y \sum_{i=m+1}^{d} \alpha_i e_i \cdot \sum_{i=m+1}^{d} \eta_i \gamma_i e_i > 0
$$

$$
y \sum_{i=m+1}^{d} \eta_i \alpha_i \gamma_i (e_i \cdot e_i) > 0
$$

$$
y \sum_{i=m+1}^{d} \eta_i \alpha_i \gamma_i > 0
\tag{14}
$$

Given Equation (13), Equation (14) is satisfied if any one of the following conditions holds:

$$\begin{cases} y \sum_{i=m+1}^{d} \eta_i \alpha_i \gamma_i > -y \sum_{i=1}^{m} \eta_i \alpha_i \gamma_i \geq 0 & (15) \\[4mm] y \sum_{i=m+1}^{d} \eta_i \alpha_i \gamma_i > 0 > -y \sum_{i=1}^{m} \eta_i \alpha_i \gamma_i & (16) \end{cases}$$

Equation (15) leads to:

$$y \sum_{i=1}^{m} \eta_i \alpha_i \gamma_i \leq 0 \tag{17}$$

By adding $y \sum_{i=1}^{m} \eta_i \alpha_i \gamma_i$ to Equation (16), we can derive:

$$y \sum_{i=1}^{d} \eta_i \alpha_i \gamma_i > y \sum_{i=1}^{m} \eta_i \alpha_i \gamma_i > 0 \tag{18}$$

$\square$

### A.3 Conditions for TACT to preserve a correct prediction

**Proposition 3.** *Suppose $z$ is correctly classified by the learned decision boundary $\Delta q$. The trimmed representation $\hat{z}$ obtained via TACT will still be classified correctly if either of the conditions holds:*

1. *$y(z - \hat{z})\Delta q \leq 0$, or*

2. *$y(z - \hat{z})\Delta q > 0$, and Equation (7) holds, assuming $\hat{z}$ already satisfies the Causal Preservation condition (Proposition 2).*

$$\text{sign}\left( \sum_{i=m+1}^{d} \eta_i \alpha_i \gamma_i \right) = \text{sign}\left( \sum_{i=m+1}^{d} \alpha_i \gamma_i \right) \tag{7}$$

Equation (7) requires that when classification relies only on the representations formed by the remaining PCs, the learned decision boundary makes the same prediction as the causal decision boundary. Proposition 3 also shows that if a correct prediction is made by the learned decision boundary, TACT will preserve this correctness as long as the removed part $z - \hat{z}$ contributes negatively or does not contribute to the prediction. On the other hand, when the trimmed representation $\hat{z}$ contains sufficient causal information as established in Proposition 2, the learned decision boundary is required to align directionally with the causal decision boundary defined by the remaining PCs.

The proof provided here corresponds to this corrected version.

*Proof.* As the learned decision boundary $\Delta q$ classify $z$ correctly, we have:
$$yz \cdot \Delta q > 0$$
$$y(z - \hat{z}) \cdot \Delta q + y\hat{z} \cdot \Delta q > 0 \tag{19}$$

We can rewrite $y\hat{z} \cdot \Delta q$ as:

$$\begin{aligned} y\hat{z} \cdot \Delta q &= y \sum_{i=m+1}^{d} \alpha_i e_i \cdot \sum_{i=1}^{d} \gamma_i e_i \\ &= y \sum_{i=m+1}^{d} \alpha_i e_i \cdot \left( \sum_{i=1}^{m} \gamma_i e_i + \sum_{i=m+1}^{d} \gamma_i e_i \right) \\ &= y \sum_{i=m+1}^{d} \alpha_i e_i \cdot \sum_{i=1}^{m} \gamma_i e_i + y \sum_{i=m+1}^{d} \alpha_i e_i \cdot \sum_{i=m+1}^{d} \gamma_i e_i \\ &= 0 + y \sum_{i=m+1}^{d} \alpha_i e_i \cdot \sum_{i=m+1}^{d} \gamma_i e_i \\ &= y\hat{z} \cdot \Delta \hat{q} \end{aligned} \tag{20}$$

By combining Equation (19) and Equation (20), we can derive:

$$y(z - \hat{z}) \cdot \Delta q + y\hat{z} \cdot \Delta \hat{q} > 0 \tag{21}$$

The updated prediction by TACT is correct if and only if $y\hat{z} \cdot \Delta \hat{q} > 0$. Equation (21) shows that the value of $y(z - \hat{z}) \cdot \Delta q$ needs to be considered to derive the conditions under which $y\hat{z} \cdot \Delta \hat{q} > 0$.

1. When $y(z - \hat{z}) \cdot \Delta q \leq 0$, the removed part does not positively contribute to the prediction using the learned decision boundary, together with Equation (21), we can derive:

$$y\hat{z} \cdot \Delta \hat{q} > -y(z - \hat{z}) \cdot \Delta q \geq 0 \tag{22}$$

Equation (22) suggests that $y\hat{z} \cdot \Delta \hat{q} > 0$ is always true when $y(z - \hat{z}) \cdot \Delta q \leq 0$.

2. When $y(z - \hat{z}) \cdot \Delta q > 0$, the removed part positively contributes to the prediction using the learned decision boundary. We wish to connect with the causal decision boundary to understand the conditions. Therefore, we additionally assume $\hat{z}$ satisfies the Causal Preservation condition (Proposition 2), which suggests $y\hat{z} \cdot \Delta p > 0$.

The updated prediction is correct, i.e. $y\hat{z} \cdot \Delta \hat{q} > 0$ if:

$$\text{sign}\,(y\hat{z} \cdot \Delta p) = \text{sign}\,(y\hat{z} \cdot \Delta \hat{q})$$

$$\text{sign}\left(y \sum_{i=m+1}^{d} \alpha_i e_i \cdot \sum_{i=1}^{d} \eta_i \gamma_i e_i\right) = \text{sign}\left(y \sum_{i=m+1}^{d} \alpha_i e_i \cdot \sum_{i=m+1}^{d} \gamma_i e_i\right)$$

$$\text{sign}\left(y \sum_{i=m+1}^{d} \alpha_i \eta_i \gamma_i (e_i \cdot e_i)\right) = \text{sign}\left(y \sum_{i=m+1}^{d} \alpha_i \gamma_i (e_i \cdot e_i)\right)$$

$$\text{sign}\left(y \sum_{i=m+1}^{d} \alpha_i \eta_i \gamma_i\right) = \text{sign}\left(y \sum_{i=m+1}^{d} \alpha_i \gamma_i\right)$$

$$\text{sign}\left(\sum_{i=m+1}^{d} \eta_i \alpha_i \gamma_i\right) = \text{sign}\left(\sum_{i=m+1}^{d} \alpha_i \gamma_i\right) \tag{23}$$

$\square$

## B  Data Augmentation for CivilComments

CivilComments considers the following demographics mentioned in a comment: male, female, LGBTQ, Christian, Muslim, other religions, Black, White. We group the demographics into gender (male/female), sexuality (LGBTQ), religion (Christian/Muslim/other religions), and race (Black/White). We notice that the comments tend to mention only one of the demographics in each group, and some comments mention more than one group. To vary demographics, we propose to introduce new demographics to the comments.

We propose to randomly insert a sentence before or after the comment. The sentences being inserted are randomly drawn from a set of sentences. Each sentence in the set mentions all demographics in one of the groups. The sentences are not toxic, so they would not affect the toxicity rating of the comment. Toxic comments remain toxic, and non-toxic comments remain non-toxic when the sentence is added. We ask ChatGPT via the web interface (https://chatgpt.com) to generate 20 sentences for each demographic group. We list the sentences below. Sentences from all groups make up the set from which we randomly sample for augmentation.

**Gender(male/female)**

- "This is a post about females and males."
- "The discussion focuses on women and men."
- "Females and males are the central topic here."
- "Women and men both contribute to this conversation."

- "This explores perspectives of females and males."
- "The post highlights contributions of women and men."
- "Both females and males are part of the narrative."
- "Women and men play essential roles in this story."
- "Females and males are equally represented here."
- "This covers aspects of both women and men."
- "This is a post about women and men."
- "The discussion centers on ladies and gentlemen."
- "Females and males are the key focus here."
- "Girls and boys both play significant roles."
- "Both genders are part of this discussion."
- "This highlights contributions from men and women."
- "Ladies and gentlemen are represented here equally."
- "The focus is on both sexes and their roles."
- "Womenfolk and menfolk shape this narrative."
- "Both males and females are included in this topic."

## Sexuality (LGBTQ)

- "This is a post about LGBTQ+ and heterosexual individuals."
- "The discussion focuses on sexual minorities and heterosexual communities."
- "This highlights experiences of both LGBTQ+ and cisgender people."
- "The post compares queer and non-queer perspectives."
- "This covers topics relevant to both LGBTQ+ and straight groups."
- "Gender-diverse and cisgender voices are included in this conversation."
- "The focus is on LGBTQ+ and heterosexual rights and issues."
- "Both sexual minorities and heterosexual people's experiences are addressed here."
- "This post examines the lives of gender-nonconforming and cisgender individuals."
- "The post explores the intersection of queer and non-queer identities."
- "LGBTQ+ and heterosexual people both contribute to this topic."
- "This content engages with both gender-diverse and cisgender communities."
- "The article offers insights into the experiences of LGBTQ+ and non-LGBTQ+ individuals."
- "This is a post about LGBTQ+ and heterosexual experiences in society."
- "Both sexual minorities and heterosexual groups have a place in this discussion."
- "This conversation includes both LGBTQ+ and cisgender perspectives."
- "We explore issues affecting both sexual minorities and heterosexual individuals."
- "This is about the relationships between LGBTQ+ and heterosexual people."
- "The focus is on creating unity between LGBTQ+ and cisgender communities."
- "This post discusses challenges faced by both gender-diverse and cisgender people."

## Religion (Christian/Muslim/other religions)

- "This is a post about Christians, Muslims, and followers of other faiths."
- "The discussion focuses on Christians, Muslims, and practitioners of different religions."
- "This highlights the experiences of Christians, Muslims, and believers from various traditions."
- "The post compares Christian, Muslim, and other spiritual practices."
- "This covers topics relevant to Christians, Muslims, and people of other religious backgrounds."
- "The voices of Christians, Muslims, and adherents of different faiths are included in this conversation."

- "The focus is on Christian, Muslim, and interfaith perspectives."
- "Both Christians, Muslims, and people of other beliefs contribute to this discussion."
- "This post examines the lives of Christians, Muslims, and followers of other religions."
- "The post explores the intersection of Christianity, Islam, and other spiritual practices."
- "Christians, Muslims, and people from diverse faiths share common values of compassion."
- "This content engages with Christians, Muslims, and those from various religious traditions."
- "The article offers insights into the teachings of Christians, Muslims, and other faith communities."
- "This is a post about Christians, Muslims, and adherents of various world religions."
- "Both Christians, Muslims, and individuals from different belief systems are included in this conversation."
- "The focus is on how Christians, Muslims, and people of other religions practice faith."
- "This conversation includes insights from Christians, Muslims, and followers of other spiritual paths."
- "We'll explore issues affecting Christians, Muslims, and people from various religious backgrounds."
- "This is about the relationships between Christians, Muslims, and those of other beliefs."
- "The post discusses shared values between Christians, Muslims, and adherents of other religions."

### Race (Black/White)

- "This is a post about Black and White communities."
- "The discussion focuses on African American and Caucasian experiences."
- "This highlights the perspectives of Black and White individuals."
- "The post compares the lives of Black and White people."
- "This covers topics relevant to both Black and White races."
- "The voices of African Americans and Caucasians are included in this conversation."
- "The focus is on Black and White racial dynamics."
- "Both Black and White communities contribute to this discussion."
- "This post examines the experiences of Black and White individuals."
- "The post explores the intersection of African American and European American identities."
- "Black and White people play vital roles in shaping society."
- "This content engages with the experiences of Black and White groups."
- "The article offers insights into the lives of Black and White people in different settings."
- "This is a post about African American and White American experiences."
- "Both Black and White cultures have unique contributions to the world."
- "The focus is on both Black and White perspectives in social issues."
- "This conversation includes both Black and White voices."
- "We'll explore the relationship between Black and White individuals."
- "This is about the interactions between African Americans and Caucasians."
- "The post discusses challenges faced by both Black and White communities."

## C  Augmentation Design and Selection

Data augmentation requires careful consideration in order to achieve strong performance. It should heuristically maximize variations along non-causal directions and minimize variations along causal directions, so that the directions corresponding to non-causal features are well identified by Principal Component Analysis.

In practice, the augmentation can be treated as a hyperparameter to search over. The data collection process that raises variation and features that affect the prediction target should be analyzed to propose a set of augmentations that are semantically invariant with respect to the prediction target, yet introduce variability in other, non-causal aspects.

For example, for the commonly studied image classification task, we recommend searching over general image augmentations, such as AutoAugment [6] and RandomAugment [7]. These augmentations preserve the critical causal features, particularly the shape information of objects [10], while simultaneously injecting variability into less essential aspects. Our experiments examine the effect of different augmentation strategies on datasets where images serve as the predictive input. As shown in Table 3, augmentation affects model performance, but AutoAugment and RandomAugment could provide consistent improvements over no adaptation.

The most effective way to select the augmentation is to test on a small subset of labeled test data.

Table 3: Performance of TACT with different augmentation strategies.

| Augmentation | Birdcalls | Camelyon17[1] | ImageNet-R | ImageNet-V2 |
|---|---|---|---|---|
| no TTA | 22.74 | 62.31 | 41.83 | 62.97 |
| Stain color jitter/color jitter | 31.14±1.69 | 70.17±0.05 | 41.78±0.01 | 61.88±0.11 |
| AutoAugment | 27.61±2.25 | 72.04±0.12 | 43.29±0.07 | 63.33±0.10 |
| RandomAugment | 32.19±1.26 | 79.71±0.07 | 43.59±0.02 | 62.99±0.10 |

# D   Details of Test-Time Adaptation Experiment

## D.1   Model Used for Adaptation

For Birdcalls and Camelyon, to our knowledge, there were no publicly available ViT-B/32 models trained on the datasets. Therefore, we train a model using the standard empirical risk minimization. The training scripts and models can be found at our code repository `https://github.com/NancyQuris/TACT`. The details of the training are:

- Birdcalls uses a batch size of 16 and is trained for 100 epochs. AdamW is employed as the optimizer, with a learning rate of 5e-5 and weight decay of 0.001. As specified in [9], the training starts from a weight pretrained on ImageNet, and the best model is selected by macro F1 on the in-distribution validation split.
- Camelyon17 uses a batch size of 32 and is trained for 30 epochs. SGD is employed as the optimizer, with a learning rate of 5e-5 and momentum 0.9. As instructed in [28], the training starts from a randomly initialized weight, and the best model is selected by the average classification accuracy on the validation domain.

For CivilComments, we use the model provided by Wilds [28]. The model was trained on the training domain of CivilComments using empirical risk minimization. The model can be found in `https://worksheets.codalab.org/rest/bundles/0x17807ae09e364ec3b2680d71ca3d9623/contents/blob/best_model.pth`.

For ImageNet-R and ImageNet-V2, we use the model published by torchvision. The model was trained on ImageNet using empirical risk minimization. The pretrained weight `ViT_B_32_Weights.IMAGENET1K_V1` is loaded to the model for test-time adaptation.

## D.2   Hyperparameter Search Space

We perform a grid search to find the best hyperparameters for the baseline methods we compared with. For backpropagation-free methods, here list the details of the hyperparameters searched:

---

[1]The performance of AutoAugment and RandomAugment on Camelyon17 is under the removal of principal components beginning with the 2nd. We observe that removing the first principal component only results in performance degradation. We hypothesize that important causal features might be present in the first principal component.

- T3A: Following [19], $M$, the number of representations stored to compute the centroid of each class is searched in $\{1,5,20,50,100, N/A\}$, where N/A means storing all representations.
- LAME: Following [4], the $k$ used in $k$-nearest neighbours is searched in $\{1,3,5\}$, and the kernel to compute distance is searched in $\{kNN, linear, rbf\}$.
- FOA: Following [36], we use 3 prompts. The population size is set to $4 + 3 \times \log(\text{prompt dim})$. The $\lambda$ to balance entropy and representation distance is searched in $\{0.2, 0.4\}$.

For all backpropagation-based methods, we search the learning rate in $\{1e\text{-}3, 1e\text{-}4, 1e\text{-}5, 1e\text{-}6\}$. The adaptation is performed in a non-episodic way. For other hyperparameters used in each method, the details are listed below:

- SHOT: The method was originally proposed for source-free domain adaptation [30]. Following [19] that adapts it as a TTA strategy, $\beta$, the hyperparameter to balance information maximization and cross entropy, is set to 0.1. The hyperparameter to filter confident pseudo-labels is set to 0.9. Adam is used as the optimizer. The feature extractor is updated during adaptation. The adaptation step is set to 1.
- Tent: Following [51], SGD is used as the optimizer with momentum 0.9. The affine parameters of normalization layers are updated during adaptation. The adaptation step is set to 1.
- SAR: Following [38], the margin $E_0$ is set to $0.4 \times \ln C$, where $C$ is the number of classes. To recover the model, the moving average factor is set to 0.9, and the reset constant is set to 0.2. SGD is used as the base optimizer with sharpness-aware minimization (SAM). The momentum for SGD is set to 0.9. $\rho$ in SAM is set to 0.05. The affine parameters of shallow normalization layers are updated. Normalization layers in the $9^{th}$-$11^{th}$ block in the feature extractor are frozen during adaptation. The adaptation step is set to 1.
- DeYO: Following [29], we search over the three augmentations $\{$patch shuffling, pixel shuffling, occlusion$\}$ to destory causal features. The patch size in patch shuffling is set to 4. For occlusion, the occlusion size is set to $(H/2) \times (W/2)$, where $H$ and $W$ stand for the height and width of the image. The occulsion starts from $(H/4)^{th}$ row and $(W/4)^{th}$ column. The DeYO margin is set to $0.5 \times \ln C$, and the margin $E_0$ is set to $0.4 \times \ln C$, where $C$ is the number of classes. The PLPD threshold is searched in $\{0.2, 0.3, 0.5\}$. SGD is used as the optimizer with momentum 0.9. The affine parameters of shallow normalization layers are updated. Normalization layers in the $9^{th}$-$11^{th}$ block in the feature extractor are frozen during adaptation. The adaptation step is set to 1.
- TAST: Following [21], we search the number of nearby support examples $N_s$ in $\{1, 2, 4, 8\}$. $M$, the number of support examples per class is searched in $\{1,5,20,50,100, N/A\}$, where N/A means storing all representations. The number of adaptation modules $N_e$ is set to 20. Adam is used as the optimizer. The trainable module added on top of the feature extractor is adapted. The adaptation step is searched in $\{1, 3\}$.
- TSD: Following [53], the hyperparameter for feature filter $M$ is searched in $\{1, 5, 20, 50, 100, N/A\}$, where N/A denotes no entropy filter. The tradeoff parameter $\lambda$ to balance TSD loss and MSLC loss is set to 0.1. Adam is used as the optimizer. Adapting $\{$affine parameters, classifier, feature extractor, all parameters$\}$ is searched. The adaptation step is set to 1.
- PASLE: Following [17], we search the the threshold in $\{0.2, 0.4, 0.6, 0.8\}$. The threshold gap is set to 0.1. The $\tau_{\text{des}}$ is searched in $\{1e\text{-}3, 1e\text{-}4\}$. The buffer size is set to 16, 1/4 of the batch size we used. Adam is used as the optimizer. Adapting $\{$affine parameters, classifier, feature extractor, all parameters$\}$ is searched. The adaptation step is set to 1.

### D.3   Hardware and Software Used

We perform experiments on the NVIDIA V100 GPU with 32GB memory. When the batch size is set to 64, the memory of 1 GPU is sufficient to perform test-time adaptation using TACT as well as all the baseline methods.

We implement TACT using PyTorch 2.1.2. Singular vector decomposition implemented by `torch.linalg.svd()` is used to compute the principal components, as it is computationally more stable than spectral decomposition. Since the covariance matrix is a symmetric positive semi-definite matrix, the singular vectors are the same as the eigenvectors.

# E Additional Performance Study

## E.1 TTA Performance on Larger Models

We examine TACT's effectiveness on larger models, specifically ViT-B/16 for images and BERT for texts. The experiment setup is consistent with that described in Section 6. Table 4 presents the performance of TACT and other state-of-the-art backpropagation-free methods on the larger architectures. Across all datasets except ImageNet-R, TACT achieves the best performance, ranking second on ImageNet-R. These results demonstrate the scalability of TACT to larger models.

The models for Birdcalls and Camelyon are trained under the same setting as that for ViT-B/32 stated in Appendix D.1. We follow the guidance of CivilComments' publisher to train BERT. The models we trained are included in our code repository. ViT-B/16 backbone for ImageNet-R and ImageNet-V2 is published by torchvision.

Table 4: Test-time adaptation performance of backpropagation-free methods on larger models. The best performance of each dataset is in bold.

| Method | Birdcalls | Camelyon17 | CivilComments | ImageNet-R | ImageNet-V2 |
|--------|-----------|------------|---------------|------------|-------------|
| No TTA | 27.10 | 65.37 | 67.62 | 44.06 | 69.57 |
| T3A | 28.32±1.60 | 72.72±0.73 | 67.46±0.00 | 43.99±0.08 | 69.67±0.04 |
| LAME | 27.48±1.44 | 68.50±0.11 | 67.65±0.04 | 44.04±0.04 | 69.59±0.01 |
| FOA | 27.89±0.54 | 67.15±0.67 | - | **47.53±2.73** | 69.68±0.04 |
| TACT | **33.65±2.11** | **72.85±0.02** | **69.76±0.44** | 45.59±0.01 | **69.71±0.02** |

## E.2 Synergy with Training-time Augmentation

The "no TTA" baselines of BirdCalls, Camelyon17, and CivilComments are trained without the augmentations used by TACT to identify and reduce non-causal features. To assess TACT's synergy with training-time augmentation, we trained models using the same augmentations as those applied by TACT and then performed test-time adaptation. For ImageNet-R and ImageNet-V2, the "no TTA" baseline provided by torchvision was trained with AutoAugment using the ImageNet policy.

Table 5 shows the test-time adaptation performance of TACT on models trained with the same augmentation strategy. The results show that, even when models are trained with these augmentations, TACT further improves test-time performance. This highlights TACT's ability to synergize with training-time augmentation and provides strong evidence of its effectiveness and generalizability.

Table 5: Test-time adaptation performance of TACT with training-time augmentation models.

| | Birdcalls | Camelyon17 | CivilComments | ImageNet-R | ImageNet-V2 |
|--------|-----------|------------|---------------|------------|-------------|
| no TTA (train time aug) | 29.86 | 74.09 | 64.60 | 41.83 | 62.97 |
| + TACT | 30.57±0.96 | 77.27±0.03 | 68.84±0.20 | 43.29±0.07 | 63.33±0.10 |

## E.3 TTA Performance under Different Batch Size

We study the test-time adaptation performance of TACT on ImageNet-R when the test batch size varies. Table 6 shows the result when the test batch size is set to 1, 4, 16, 64 and 128, respectively. The performance remains stable across different batch sizes. Even with a batch size of 1, the performance only decreases by 0.06% compared to a batch size of 64. Moreover, TACT still improves performance by 1.7% over the no-adaptation baseline when only one sample is available per batch during adaptation. The result suggests that TACT is robust to variations in batch size, maintaining high performance even when batch sizes are small. This makes it well-suited for situations where the number of test samples per batch is constrained.

Table 6: Test-time adaptation performance (%) of TACT on ImageNet-R under different batch sizes.

| no TTA | batch size = 1 | batch size = 4 | batch size = 16 | batch size = 64 | batch size = 128 |
|---|---|---|---|---|---|
| 41.83 | 43.53±0.02 | 43.51±0.03 | 43.55±0.06 | 43.59±0.02 | 43.56±0.03 |

### E.4    Computational Cost

We compared the computational requirements of TACT with those of other backpropagation-free methods on the Birdcalls dataset using a ViT-b/32 backbone. As shown in Table 7, TACT incurs higher time and GPU memory consumption relative to alternative approaches. Nevertheless, this additional computational cost results in substantial performance gains (Table 1), which justifies the trade-off. Future work may explore optimization strategies, such as more efficient eigendecomposition techniques for PCA, to reduce the overhead.

Table 7: Time and GPU memory required by backpropagation-free methods on Bridcalls.

|  | time (second) | GPU memory (MB) |
|---|---|---|
| T3A | 7.67 | 667.42 |
| LAME | 7.34 | 667.42 |
| FOA | 16.83 | 667.42 |
| TACT (num aug=128) | 112.22 | 1750.21 |
| TACT (num aug=256) | 170.00 | 2966.21 |
| TACT (num aug=512) | 323.62 | 5398.21 |

### E.5    Additional Visualization of Predictions after Causal Trimming

We provide more GradCAM visualization of the original predictions and the predictions made by TACT on samples from ImageNet-R. Figure 5 shows the visualizations.

Compared to original predictions, predictions made by TACT focus less on non-causal information. For example, TACT pays less attention to the background of the warplane example, and the blowfish example. The focus on the information that is semantically correlated with the class is retained in predictions made by TACT in the above examples. When the causal information is not important to the original prediction, prediction made by TACT leverages the causal information and thus turn the wrong prediction correct, as shown in the example of jellyfish and bloodhound.

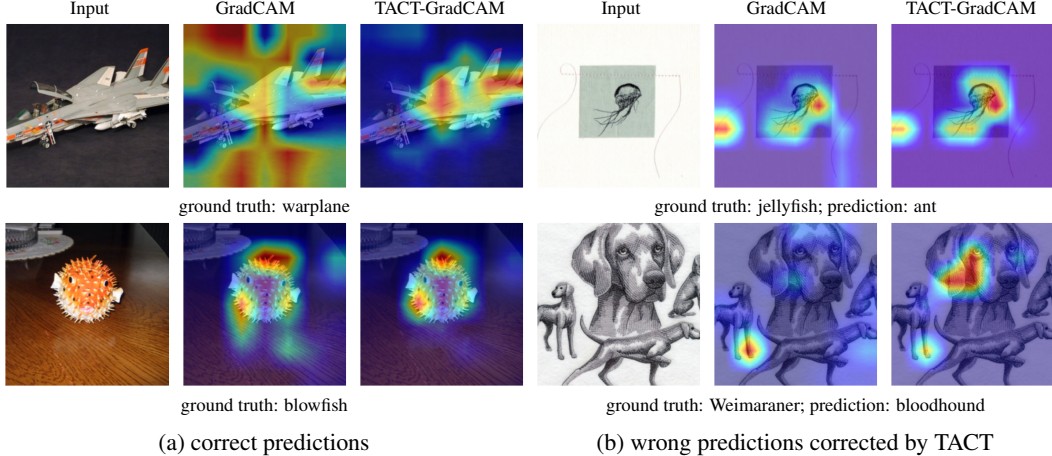

(a) correct predictions          (b) wrong predictions corrected by TACT

Figure 5: Additional GradCAM visualizations of the original predictions and TACT's predictions.

# F  Alternative Design of TACT

## F.1  ICA to Find Non-Causal Directions

We experiment using an alternative direction finding method, Independent Component Analysis (ICA) with TACT. We rank the independent components by the variance of the scalars of features on the components. We remove the top independent components that have maximum variance. Table 8 shows the result on the Birdcalls dataset. ICA performs inferior to Principal Component Analysis (PCA), but better than no adaptation. Although ICA overcomes the orthogonality constraints of PCA, it only looks for statistically independent components and assumes each component follows a non-Gaussian distribution. Causal and non-causal features might not follow the non-Gaussian distribution assumption under augmentations that vary non-causal features.

Table 8: Performance of TACT with ICA to find non-causal directions.

| no TTA | TACT w/ PCA | TACT w/ ICA |
|--------|-------------|-------------|
| 22.74  | 31.14±1.69  | 25.53±1.06  |

## F.2  Causal Trimming Based on a Threshold

We consider using the variance that the top principal components (PC) account for as a threshold to decide whether causal trimming is conducted or not. When the augmentation only changes non-causal features and causal features remain unchanged, datapoints that are invariant to augmentations should have smaller variance of the top PCs. Thus, if the variance is smaller than a threshold, causal trimmings will not be conducted on the data. As the range of variance is not known and it could change significantly, setting a numerical threshold might not be feasible. We consider normalized variance, where we divide the variance of top PCs by the sum of variances of all PCs. Table 9 shows the result on the Birdcalls dataset. Removing components based on a threshold does not outperform using no threshold.

Table 9: Performance of TACT when causal trimming is performed based on a threshold $\tau$.

| no TTA | TACT | TACT ($\tau$=0.1) | TACT ($\tau$=0.2) | TACT ($\tau$=0.3) |
|--------|------|-------------------|-------------------|-------------------|
| 22.74  | 31.14±1.69 | 30.99±2.18 | 31.03±2.19 | 28.03±3.12 |

