# OpenReview forum: "Test-Time Adaptation by Causal Trimming"
_NeurIPS.cc/2025/Conference — NeurIPS 2025 poster_

### Official Review · Reviewer_DW91 · 2025-07-01

**Clarity:** 3
**Significance:** 2
**Originality:** 2
**Rating:** 3
**Confidence:** 4

**Summary:**

The paper proposes a method for test-time adaptation by removing the non-causal components from intermediate representations. At test-time augmentations are applied and PCA is used to remove the components with the highest variance. Experiments were performed on a range of datasets, and supportive results were shown compared to the baselines.

**Questions:**

- Was the “no TTA” baseline trained with the same augmentations used for TACT?
- What would be the performance of the method if the hyperparameters were tuned on validation data instead?
- Have the authors tried removing components based on a threshold instead? If the datapoint is invariant to the augmentation, will this remove the “causal” components too?

**Ethical Concerns:**

["NO or VERY MINOR ethics concerns only"]

**Final Justification:**

I appreciate the authors rebuttal and the additional results, they were insightful. However, my concern about the choice of augmentation stills stands. While they work better than no TTA, the standard augmentations that works the best for one dataset, results in the worst performance for another dataset. Thus, I will be keeping my score.

**Limitations:**

yes

**Quality:**

2

**Strengths And Weaknesses:**

**Strengths**

- The paper was well written and easy to follow.
- The experiments were done on a diverse datasets and domains (from audio signals, text to images) with supportive results.

**Weaknesses**

- The selected augmentations seems to assume some knowledge of about the distribution shift, i.e., they are selected to be similar to the shift for each dataset. Was the no TTA baseline trained with these augmentations? If not, It would be useful to see if learning invariances via training time augmentation alone is sufficient.
- Performance of the method in practice. The paper mentions that oracle selection to used to select the hyperparameters (nb of augmentations, components), allowing for fair comparison across methods. While this can give a fair comparison, it makes it hard to understand the performance of the method in practice. From Fig. 4, there can be large differences in performance.
- Computation cost. Since the method performs PCA, it can be more expensive that the baselines. Including the computational cost of each method make it easy to assess the trade-offs and make a more informed decision for time sensitive applications.

---

> ### Author Rebuttal · Authors · 2025-07-31
>
> Thank you for the extensive feedback and for recognizing our empirical analysis.
> We discuss the weakness and questions raised in the review below.
>
> ### Weakness
>
> **W1. "no TTA'' baseline and training time augmentation**
>
> The "no TTA'' baseline of BirdCalls, Camelyon17, and CivilComments were not trained with the augmentations used by TACT to identify and reduce non-causal features.
> Following the instructions of the data publisher, no data augmentation was used when training the baseline with empirical risk minimization.
> To understand the effectiveness of training time augmentation, we trained the models with the data augmentations used by TACT. The result is shown in the table below.
> Training time augmentation is not always sufficient, as it achieves worse performance than TACT on BirdCalls and Camelyon17.
> We further show that, when training time augmentation is performed, TACT can still improve the performance.
> These results suggest TACT's usefulness when tackling distribution shift.
>
> |                         | Birdcalls      | Camelyon17     | CivilComments  |
> |-------------------------|----------------|----------------|----------------|
> | no TTA                  | 22.74          | 62.31          | 55.38          |
> | + TACT                  | 29.99$\pm$0.80 | 70.17$\pm$0.05 | 71.80$\pm$0.35 |
> | no TTA w/train time aug | 29.86          | 74.09          | 64.60          |
> | + TACT                  | 30.30$\pm$0.27 | 77.27$\pm$0.03 | 68.84$\pm$0.20 |
>
> For ImageNet-R and ImageNet-V2, the "no TTA'' baseline provided by torchvision was trained with AutoAugment using the ImageNet policy. The table below shows the performance of TACT on ImageNet-R and ImageNet-V2 with the same augmentation. TACT improves the performance.
>
> |             | ImageNet-R     | ImageNet-V2    |
> |-------------|----------------|----------------|
> | no TTA      | 41.83          | 62.97          |
> | AutoAugment | 43.29$\pm$0.07 | 63.33$\pm$0.10 |
>
> **W2.  Performance selected by validation data**
>
> For Birdcalls, Camelyon, and CivilComments, we use the validation domain for model selection.
> For ImageNet-R and ImageNet-V2, we use the validation set of ImageNet for model selection.
> The performance is shown in the table below.
> TACT performs the best on all datasets except for Birdcalls. For Birdcalls, TACT performs the second best.
>
> | Method | Birdcalls               | Camelyon17              | CivilComments           | ImageNet-R              | ImageNet-V2             |
> |--------|-------------------------|-------------------------|-------------------------|-------------------------|-------------------------|
> | No TTA | 22.74                   | 62.31                   | 55.38                   | 41.83                   | 62.97                   |
> | T3A    | 23.71$\pm$1.74          | 68.54$\pm$0.06          | 56.43$\pm$0.00          | 41.66$\pm$0.04          | 62.93$\pm$0.02          |
> | LAME   | 22.81$\pm$0.71          | 62.38$\pm$0.03          | 56.00$\pm$0.10          | 41.71$\pm$0.04          | 63.00$\pm$0.02          |
> | FOA    | **26.72$\pm$0.70** | 58.36$\pm$0.77          | -                       | 41.46$\pm$0.16          | 62.75$\pm$0.09          |
> | TACT   | 25.22$\pm$1.19          | **70.17$\pm$0.05** | **71.33$\pm$0.36** | **42.10$\pm$0.04** | **63.02$\pm$0.12** |
>
> **W3. Computation cost**
>
> We compared the computation cost of TACT with other backpropagation-free methods on Birdcalls. The table below shows that TACT requires more time and memory than other backpropagation-free methods.
> However, this increase in computational cost leads to a significant improvement in performance shown in Table 1 of the main paper, which we believe justifies the trade-off.
> In many practical applications, the improved performance is critical and outweighs the added cost.
> Further optimization techniques on efficient eigendecomposition for PCA could be applied to reduce the overhead if needed, which we leave as future work.
>
> | Method | time (second) | GPU memory (MB) |
> |--------------------|---------------|-----------------|
> | T3A                | 7.67          | 667.42          |
> | LAME               | 7.34          | 667.42          |
> | FOA                | 16.83         | 667.42          |
> | TACT (num aug=128) | 112.22        | 1750.21         |
> | TACT (num aug=256) | 170.00        | 2966.21         |
> | TACT (num aug=512) | 323.62        | 5398.21         |
>
> ### Questions
> **Q1. ''no TTA" baseline**
>
> Please refer to W1.
>
> **Q2. Performance selected by validation data**
>
> Please refer to W2.
>
> **Q3. Removing components based on a threshold**
>
> We consider using the variance that the top PC accounts for as a threshold ($\tau$) to decide whether causal trimming is conducted or not.
> When the augmentation only changes non-causal features and causal features remain unchanged, datapoints that are invariant to augmentations should have smaller variance of the top PCs.
> Thus, if the variance is smaller than a threshold, causal trimmings will not be conducted on the data.
> As the range of variance is not known and it could change significantly, setting a numerical threshold might not be feasible.
> We consider normalized variance, where we divide the variance of top PCs by the sum of variances of all PCs.
>
> The table below shows  the performance. Current experiments on Birdcall with threhsold does not improve the performance.
> Future work can study this issue in more details.
>
> |           | no TTA | TACT           | TACT ($\tau$=0.1) | TACT ($\tau$=0.2) | TACT ($\tau$=0.3) |
> |-----------|--------|----------------|-------------------|-------------------|-------------------|
> | Birdcalls | 22.74  | 29.99$\pm$0.80 | 28.95$\pm$0.92    | 29.03$\pm$1.17    | 25.69$\pm$1.57    |

---

> ### Author Response · Authors · 2025-08-06
>
> Dear Reviewer DW91,
>
> Thank you so much for the feedback you provided.
> We hope that our reply addresses your concerns.
> Please let us know if you have any further concerns or suggestions—we're more than happy to discuss them.

---

> > ### Comment · Reviewer_DW91 · 2025-08-07
> >
> > Thank you for the clarifications and for the additional experiments. The results were insightful and it is good to see the improvements even when selecting using validation data. The method is indeed more expensive but these trade-offs can be made by the user.
> >
> > However, the main issue, as also raised by the other reviewers is that the augmentations need to be carefully selected. Furthermore, the new results of no TTA w/train-time aug shows that it does give a boost in performance, although still less than that of TACT while being much more straightforward to implement. I appreciate the additional results but I am still inclined to keep my score.

---

> > > ### Author Response · Authors · 2025-08-07
> > >
> > > Dear Reviewer DW91,
> > >
> > > Thank you again for your feedback. Here is our discussion on the additional raised concerns:
> > >
> > > **Selection of augmentation**
> > >
> > > TACT could still be robust when the augmentations used can vary the non-causal features and keep the causal features approximately invariant.
> > > In fact, augmentations that maintain key causal features could be helpful.  For example, for the image classification task, general image augmentations such as AutoAugment and RandomAugment generally work well with TACT.
> > > Our experiment of using them with Birdcalls and Camelyon shows that they can also offer performance gain compared to no adaptation.
> > >
> > > | Augmentation                    | Birdcalls      | Camelyon17     |
> > > |---------------------------------|----------------|----------------|
> > > | no TTA                          | 22.74          | 62.31          |
> > > | Stain color jitter/color jitter | 29.99$\pm$0.80 | 70.17$\pm$0.05 |
> > > | AutoAugment                     | 27.45$\pm$1.28 | 66.42$\pm$0.01 |
> > > | RandomAugment                   | 31.76$\pm$0.19 | 65.60$\pm$0.08 |
> > >
> > >
> > > The paper wants to highlight that reducing non-causal features by adjusting the learned representations is helpful for TTA. Data augmentation is one approach to identify non-causal features. In practice, the augmentation can be treated as a hyperparameter to search over.
> > > Alternative to data augmentation, if other information, such as a small set of labeled data from multiple distributions, is given, non-causal features can also be identified by comparing the representations of data with the same labels, which we leave for future work.
> > >
> > > **no TTA w/ train-time aug**
> > >
> > > The test-time adaptation offered by TACT represents efforts to solve distribution shift at test time, which are complementary to the efforts achieved during training.
> > > We demonstrate in W1 of our rebuttal that TACT can further enhance performance on models trained with training-time augmentation.
> > > We hope the additional results on TACT being able to synergize and improve over training-time augmentation can be treated as supporting evidence for broader generalizability rather than a limitation.
> > >
> > >
> > > We hope that our reply is helpful, and we are looking forward to your reply.

---

### Official Review · Reviewer_QQc2 · 2025-07-02

**Clarity:** 3
**Significance:** 3
**Originality:** 4
**Rating:** 5
**Confidence:** 3

**Summary:**

This paper proposes TACT (Test-time Adaptation by Causal Trimming), a novel backpropagation-free method for test-time adaptation (TTA) under distribution shifts. The key insight is that performance degradation often stems from reliance on non-causal features. TACT addresses this by identifying and removing these non-causal components in the representation space using PCA on augmentations that vary non-causal attributes while preserving causal semantics.

Specifically, TACT:

* Uses targeted data augmentations to induce variability in non-causal dimensions.

* Applies PCA to extract directions of maximum variance (assumed to align with non-causal features).

* Removes projections onto these principal components (PCs) from both the sample representation and class prototypes.

* Uses a moving average over the trimmed prototypes for stable prediction.

* Provides theoretical guarantees for when causal trimming helps or preserves performance.

* Demonstrates strong empirical results across five diverse datasets, showing consistent improvements over state-of-the-art (SOTA) backpropagation-free and backpropagation-based TTA methods.

**Questions:**

* Your method assumes that causal features remain invariant under augmentations while non-causal ones vary. How robust is TACT when this assumption is only approximately satisfied?

* PCA assumes that variance-aligned directions are linearly separable. But in high-capacity models like ViT or BERT, representation entanglement is common. How sensitive is TACT to this? Have you considered using nonlinear alternatives (e.g., kernel PCA or ICA)?

* The number of removed PCs (m) appears to significantly influence performance (Figure 4). Can you elaborate on how this is tuned in a realistic, non-oracle setting?

**Ethical Concerns:**

["NO or VERY MINOR ethics concerns only"]

**Final Justification:**

The rebuttal satisfactorily addresses key concerns, showing that generic augmentations (e.g., AutoAugment, RandomAugment) work across datasets and that TACT remains effective under approximate disentanglement and with large models. The PCA reliance is acknowledged, with empirical results suggesting robustness despite potential entanglement. Guidance on choosing *m* is provided, though still heuristic. Overall, the method is novel, well-supported theoretically and empirically, and offers a meaningful advance in backpropagation-free TTA.

**Limitations:**

yes

**Paper Formatting Concerns:**

no issues

**Quality:**

3

**Strengths And Weaknesses:**

## Quality
### Strengths:

* Solid methodological foundation combining causal assumptions with representational geometry.

* Clear theoretical propositions (with assumptions) analyzing when and why trimming helps.

* Sound and comprehensive experimental protocol using five real-world datasets with meaningful shifts.

* Ablation studies and Grad-CAM visualizations strongly support claims.

### Weaknesses:

* Performance gains rely heavily on carefully selected augmentations that preserve causal content, which may not generalize across tasks or be feasible without domain expertise.

* PCA-based trimming assumes non-causal features dominate variance directions, which might not always hold, especially under entangled representation spaces.

## Clarity
### Strengths:

* The writing is clear, structured, and technically precise.

* Figures (especially Grad-CAM and ablations) help illustrate key points.

* Theoretical section is accessible and intuitive despite formal proofs being deferred to appendix.

### Weaknesses:

* The role of the hyperparameter m (number of trimmed PCs) is critical but tuning guidance is heuristic and empirical.

* The limitations section could more thoroughly discuss failure cases (e.g., when causal features are not orthogonal to spurious ones).

## Significance
### Strengths:

* Offers a promising new direction in test-time adaptation that is backpropagation-free yet effective.

* Bridges causal reasoning with representation learning and adaptation  a direction of growing interest.


## Originality
### Strengths:

* The idea of causal trimming based on PCA over augmentations is novel in TTA.

* Synthesizes ideas from disentanglement, causality, and test-time learning in a unique way.

* Extends beyond label-free entropy minimization or pseudo-label refinement by directly adjusting the representation space.

### Weaknesses:

Some ideas, such as augmenting data to identify spurious directions, have been explored in domain generalization and DeYO. However, TACT’s trimming strategy is distinct.

---

> ### Author Rebuttal · Authors · 2025-07-31
>
> Thank you for the comprehensive feedback and for praising the proposed method as well as our empirical and theoretical analysis. A detailed discussion of the concerns and questions is presented below.
> ### Weakness
>
> **Quality-W1. Reliance on augmentation and generalizability**
>
> We agree that augmentations need to be carefully selected to achieve good performance.
> In practice, the augmentation can be treated as a hyperparameter to search over.
> Our basic assumption is that knowledge about the prediction target is known, based on which augmentations that are semantically invariant with respect to the prediction target can be designed.
> For example, for the commonly studied image classification task, we recommend searching over general image augmentations, such as AutoAugment and RandomAugment. These augmentations preserve the key causal features, particularly the shape information of objects [1].  Our experiments in the main paper of using them with ImageNet-R and ImageNet-V2 show state-of-the-art performance.
> Our experiment of using AutoAugment and RandomAugment with Birdcalls and Camelyon shows that they can also offer performance gain compared to no adaptation, suggesting the potential generalization ability of TACT.
>
> | Augmentation                    | Birdcalls               | Camelyon17              |
> |---------------------------------|-------------------------|-------------------------|
> | no TTA                          | 22.74                   | 62.31                   |
> | Stain color jitter/color jitter | 29.99$\pm$0.80          | 70.17$\pm$0.05 |
> | AutoAugment                     | 27.45$\pm$1.28          | 66.42$\pm$0.01          |
> | RandomAugment                   | 31.76$\pm$0.19 | 65.60$\pm$0.08          |
>
> **Quality-W2. Non-causal features dominate variance directions and entangled representation space**
>
> We select data augmentation strategies to heuristically maximize variations along non-causal directions and minimize variations along causal directions, which leads to improved disentanglement after applying PCA. Therefore, it is critical to select the augmentations carefully.
>
> Theoretically, TACT does not rely on perfect disentanglement.
> We show that when causal features are partially trimmed,
> if sufficient causal information is preserved and the learned decision boundary is directionally aligned with the causal decision boundary, TACT would perform well.
> Empirically, in Section 6, we show that even approximate separation of causal and non-causal features achieved by PCA was sufficient to yield gains in performance and robustness across diverse datasets.
>
> **Clarity-W1. Tuning guidance of number of trimmed PCs**
>
> When the oracle is not accessible, m in general can be set to 1.
>
> **Clarity-W2. Failure cases**
>
> When causal features and spurious features are not orthogonal, removing spurious features will affect the remaining causal features, which likely also degrade performance. In the extreme, feature removal will not work if spurious features are perfectly correlated with causal features in the training data. This also corresponds to our theoretical analysis where sufficient causal features need to be preserved to make correct predictions.
>
> ### Questions
>
> **Q1. Robustness when the assumption is only approximately satisfied**
>
> Theoretically, Propositions 2 and 3 suggest that when causal features vary, resulting in the trimming of causal features, sufficient causal information needs to be preserved for TACT to perform well.
>
> Empirically, in our experiment with ImageNet-R and ImageNet-V2, we only tried to maintain the key causal features, which are object structure and shape. Other causal features that could be helpful in inferring objects, such as color when inferring strawberries, are altered.
> This experiment shows that TACT is still able to improve performance when the assumption is only approximately satisfied.
>
> **Q2.1. Performance on larger models when representation entanglement is common**
>
> The table below shows the performance with larger backbones, including ViT-B/16 for images and BERT for texts. TACT achieves the best performance compared to other backpropagation-free methods on all datasets except ImageNet-R. TACT achieves the second best on ImageNet-R.
> The result suggests that TACT can be robust to changes in the representation space.
>
> | Method | Birdcalls               | Camelyon17              | CivilComments           | ImageNet-R              | ImageNet-V2             |
> |--------|-------------------------|-------------------------|-------------------------|-------------------------|-------------------------|
> | No TTA | 27.10                   | 65.37                   | 67.62                   | 44.06                   | 69.57                   |
> | T3A    | 27.50$\pm$0.27          | 72.72$\pm$0.73          | 67.46$\pm$0.00          | 43.99$\pm$0.08          | 69.67$\pm$0.04          |
> | LAME   | 28.11$\pm$1.38          | 68.50$\pm$0.11          | 67.65$\pm$0.04          | 44.04$\pm$0.04          | 69.59$\pm$0.01          |
> | FOA    | 28.01$\pm$0.26          | 67.15$\pm$0.67          | -                       | **47.53$\pm$2.73** | 69.68$\pm$0.04          |
> | TACT   | **35.00$\pm$2.49** | **72.85$\pm$0.02** | **69.76$\pm$0.44** | 45.59$\pm$0.01          | **69.71$\pm$0.02** |
>
> The models for Birdcalls and Camelyon are trained under the same setting as that for ViT-B/32 used in the paper. We follow the guidance of CivilComments' publisher to train BERT. ViT-B/16 backbone for ImageNet-R and ImageNet-V2 is published by torchvision.
>
> **Q2.2. Alternative direction finding methods**
>
> Although kernelPCA and ICA overcome the limitations of PCA, they incur additional requirements.
> - kernelPCA transforms the representation space via a kernel, which needs to be selected such that causal and non-causal information are retained and better separated.
> Due to this complexity, we encourage further exploration on this in future works.
> - ICA looks for statistically independent components and assumes each component follows a non-Gaussian distribution.
> Causal and non-causal features might not follow the non-Gaussian distribution assumption under augmentations that vary non-causal features.
>
>
> We experiment with using ICA with TACT on Birdcalls.
> We rank the independent components by the variance of the scalars of features on the components. We remove the top independent components that have maximum variance.
> The experiment with ICA is searched over the same number of augmentations and number of components.
> The table below shows the result.
> ICA performs inferior to PCA, but better than no adaptation.
>
> |           | no TTA | TACT w/ PCA             | TACT w/ ICA    |
> |-----------|--------|-------------------------|----------------|
> | Birdcalls | 22.74  | **29.99$\pm$0.80** | 25.32$\pm$0.35 |
>
> **Q3. Selection of the number of removed PCs**
>
> Please refer to Clarity-W1.
>
> [1] Geirhos et al., ImageNet-trained CNNs are biased towards texture; increasing shape bias improves accuracy and robustness, ICLR 2019.

---

> ### Author Response · Authors · 2025-08-06
>
> Dear Reviewer QQc2,
>
> We thank you sincerely for the feedback. We hope that our reply resolves your questions. If there are any further questions or suggestions, please do not hesitate to let us know. We would be glad to engage in any additional discussion or clarification you require.

---

> > ### Author Response · Authors · 2025-08-08
> >
> > Dear Reviewer QQc2,
> >
> > Thank you again for your encouraging feedback on our paper. We’re glad to hear that you found the work valuable. As the discussion phase comes to a close, we would greatly appreciate it if you could share any additional questions or concerns you might have. This would allow us to respond in a timely and constructive manner.
> > If our responses have addressed your concerns, we would be deeply grateful if you would consider advocating for our paper, if you feel it is appropriate. Your support would mean a great deal to us!

---

### Official Review · Reviewer_5gmS · 2025-07-03

**Clarity:** 2
**Significance:** 2
**Originality:** 2
**Rating:** 4
**Confidence:** 3

**Summary:**

This paper introduces TACT, a backpropagation-free test-time adaptation (TTA) method that actively identifies and removes non-causal components from feature representations to improve robustness under distribution shift.
The authors provide theoretical analysis (Propositions 1–3) that formalizes when trimming corrects misclassifications and preserves causal information. Empirically, TACT is evaluated on five real-world out-of-distribution benchmarks—Birdcalls, Camelyon17, CivilComments, ImageNet-R, and ImageNet-V2—showing consistent gains over state-of-the-art backprop-free methods, and further improvements when combined with gradient-based adaptation (TACT-adapt)

**Questions:**

- In Section 4.1 (Non-Causal Identification), do you project out only the first principal component, or do you remove m components—and if so, how is m chosen for each batch？

- In Table 1, some reported standard deviations for TACT-adapt are quite large (e.g., Birdcalls ±5.68). Could you report standard deviations for all datasets and perform statistical significance tests against the strongest baselines?

- For Figure 4’s hyperparameter study, would it be possible to present heatmaps of performance over various  (n,m) combinations to visualize their interaction and identify robust operating regions?

**Ethical Concerns:**

["NO or VERY MINOR ethics concerns only"]

**Final Justification:**

The authors address scalability within their tested range, but the method’s substantially higher latency compared to other approaches may limit its applicability in low-latency settings. While further PCA optimizations are suggested, they are left as future work. Given these practical constraints, I would recommend borderline acceptance.

**Limitations:**

yes

**Paper Formatting Concerns:**

No obvious concerns

**Quality:**

2

**Strengths And Weaknesses:**

Strength

- Unlike entropy-based or self-training TTA, TACT directly suppresses non-causal features via linear trimming, addressing the root cause of distribution sensitivity.

- By avoiding gradient updates during adaptation, TACT incurs only a single PCA and a few vector operations per batch, making it attractive for latency-sensitive deployment.

- Propositions 1–3 rigorously characterize conditions for error correction, causal preservation, and non-degradation of already-correct predictions, offering clear guidance on choosing the number of components m to trim.

Weaknesses

- TACT requires domain knowledge to choose augmentations that vary only non-causal factors. An empirical study of sensitivity to augmentation choice, or an automated strategy for augmentation selection, would enhance practicality.

- PCA assumes that causal directions are orthogonal to non-causal ones. In complex tasks where causal and spurious features co-vary, this may fail. Exploring non-linear direction-finding (e.g., ICA or contrastive subspace methods) could address this limitation.

---

> ### Author Rebuttal · Authors · 2025-07-31
>
> Thank you for the detailed feedback and for acknowledging that TACT is a method that addresses the root cause of distribution shift backed by theoretical analysis.
> The weaknesses and questions raised in the review are discussed below.
>
> ### Weakness
>
> **W1.1. Augmentation sensitivity analysis**
>
> The table below shows the result. The choice of augmentation would influence performance, but we see that general augmentations such as AutoAugment and RandomAugment could provide improvements over no adaptation across datasets.  To our surprise, Birdcalls even performs better with RandomAugment than the existing color jitter.
>
> | Augmentation                    | Birdcalls      | Camelyon17     | ImageNet-R     | ImageNet-V2    |
> |---------------------------------|----------------|----------------|----------------|----------------|
> | no TTA                          | 22.74          | 62.31          | 41.83          | 62.97          |
> | Stain color jitter/color jitter | 29.99$\pm$0.80 | 70.17$\pm$0.05 | 41.78$\pm$0.01  | 61.88$\pm$0.11 |
> | AutoAugment                     | 27.45$\pm$1.28 | 66.42$\pm$0.01 | 43.29$\pm$0.07 | 63.33$\pm$0.10 |
> | RandomAugment                   | 31.76$\pm$0.19 | 65.60$\pm$0.08 | 43.59$\pm$0.02 | 62.99$\pm$0.10 |
>
> **W1.2. Automated strategy to select augmentation**
>
> In practice, the augmentation can be treated as a hyperparameter to search over.
> The data collection process that raises variation and features that affect the prediction target should be analyzed to propose a set of augmentations that are semantically invariant with respect to the prediction target, yet introduce variability in other, non-causal aspects.
> For example, for the commonly studied image classification task, we recommend searching over general image augmentations, such as AutoAugment and RandomAugment, that preserve the key causal features, particularly the shape information of objects [1].
> The most effective way to select the augmentation is to test on a small subset of labeled test data.
>
> **W2.1. Orthogonality and co-varying of causal and spurious features**
>
> TACT does not rely on perfect orthogonality or disentanglement.
> The theoretical analysis in section 5 does not rely on the orthogonality assumption.
> We show that even when causal and non-causal features are not orthogonal, and causal features are partially trimmed,
> TACT performs well as long as top PCs alone cause the prediction error, or sufficient causal information is preserved after trimming, and the learned decision boundary is directionally aligned with the causal decision boundary.
> Empirically, in Section 6, we show that even approximate separation achieved by PCA was sufficient to yield gains in performance and robustness across diverse datasets.
>
> While it is true that causal and spurious features can co-vary in the training data, it is important to note that when we apply augmentations at test time, we are trying to minimize the variation of causal features, so that the influence of non-causal features can be isolated.
>
> **W2.2. Alternative direction finding methods**
>
> We experiment with using ICA with TACT on Birdcalls.
> We rank the independent components by the variance of the scalars of features on the components. We remove the top independent components that have maximum variance.
> The experiment with ICA is searched over the same number of augmentations and number of components.
> The table below shows the result.
> ICA performs inferior to PCA, but better than no adaptation.
> We agree that ICA can overcome the orthogonality assumption. However, ICA only looks for statistically independent components and assumes each component follows a non-Gaussian distribution. Thus, ICA might not be better at identifying causal and non-causal features, which might not follow the non-Gaussian distribution assumption under augmentations that vary non-causal features.
>
> |           | no TTA | TACT w/ PCA             | TACT w/ ICA    |
> |-----------|--------|-------------------------|----------------|
> | Birdcalls | 22.74  | **29.99$\pm$0.80** | 25.32$\pm$0.35 |
>
> ### Questions
>
> **Q1. Number of PCs to project out**
>
> We explain the projection in section 4.2 (Causal Trimming to Reduce Non-Causal Feature). The top-m PCs are removed. We maintain the same m for a dataset across batches.  $m$ is a hyperparameter to be selected.
> We empirically observe that setting $m=1$, i.e., removing the top PC is frequently adequate. For datasets like ImageNet-R with more complex or layered distribution shifts, removing more PCs could further boost the performance.
>
> **Q2. Statistical significance test**
>
> To clarify, all experiments are repeated three times, and we report the mean and standard deviation in Table 1.
> We use the t-test to compute the statistical significance between the results of TACT-adapt and the strongest baselines. The p-value shows how likely the results of the two methods come from the same distribution. The table below shows the result.
> Camelyon, CivilComments, and ImageNet-V2 have a p-value smaller than 0.15.
>
> |         | Birdcalls | Camelyon17 | CivilComments | ImageNet-R | ImageNet-V2 |
> |---------|-----------|------------|---------------|------------|-------------|
> | p-value | 0.7487    | 0.1436     | 0.0000        | 0.5286     | 0.0572      |
>
> We acknowledge that the improvement on Birdcalls and ImageNet-R does not seem statistically significant, which is likely due to the small number of runs.  Therefore, we performed two additional runs to compute the result. The table below shows the updated result. P-values for the two datasets are improved. We plan to conduct more runs and update the results in the revision.
>
> |            | TACT           | Best baseline  | p-value |
> |------------|----------------|----------------|---------|
> | BirdCalls  | 30.61$\pm$5.40 | 27.94$\pm$2.19 | 0.3125  |
> | ImageNet-R | 48.84$\pm$0.14 | 48.83$\pm$0.18 | 0.4557  |
>
> **Q3. Hyperparameter and performance**
>
> Figure 4 shows the performance over various $(n, m)$ combinations.
> The colored lines denote the number of augmentations, highlighting the role of $n$, while the x-axis represents $m$, the number of removed PCs. Performance values (y-axis) are determined by the intersection of each line with the respective m value.
> Empirically, we find that values of $n \in$ {128, 256, 512} provide sufficient performance.
> For the number of removed principal components $m$, we observe that removing the top principal component, which typically captures the dominant non-causal variation, is frequently adequate.
>
> [1] Geirhos et al., ImageNet-trained CNNs are biased towards texture; increasing shape bias improves accuracy and robustness, ICLR 2019.

---

> > ### Comment · Reviewer_5gmS · 2025-08-08
> > **Response**
> >
> > Thank you for your detailed response. This partially solved my concern.
> >
> > I have one following question:
> >
> > - As your approach relies on batch-wise PCA for every test batch, have you encountered scalability or efficiency issues when applying TACT on larger-scale, high-dimensional data, or in low-latency settings?

---

> ### Author Response · Authors · 2025-08-06
>
> Dear Reviewer 5gmS,
>
> We are truly thankful for the feedback you provided.
> We hope that our reply clarifies your questions.
> If there are any additional concerns or questions you may have, please do not hesitate to let us know. We are more than willing to discuss any further suggestions or issues you might have.

---

> ### Author Response · Authors · 2025-08-08
>
> Dear Reviewer 5gmS,
>
> Thank you for raising the question regarding the scalability and efficiency of our method.
>
> We use torch's `torch.linalg.svd()` to perform PCA as discussed in our appendix. The function is able to find PCs of a batch of high-dimensional data. The internal implementation of `torch.linalg.svd()` relies on highly optimized linear algebra backends. As a result, the practical efficiency tends to scale well on modern hardware, particularly for high-dimensional data in batches commonly considered in nowadays machine learning applications.
> In our experiments, we did not encounter scalability issues. We experimented with the representation of dimension 768 and batch size up to 128. The largest dataset we tested is of size 133,782.
>
> As for latency, we compared the latency of TACT with other backpropagation-free methods on Birdcalls. The table below shows that TACT requires more time than other backpropagation-free methods.
> The increase in latency leads to a significant improvement in performance shown in Table 1 of the main paper, which we believe justifies the trade-off.
>
> |                    | time (second) |
> |--------------------|---------------|
> | T3A                | 7.67          |
> | LAME               | 7.34          |
> | FOA                | 16.83         |
> | TACT (num aug=128) | 112.22        |
> | TACT (num aug=256) | 170.00        |
> | TACT (num aug=512) | 323.62        |
>
> Further optimization techniques on efficient eigendecomposition for PCA could be applied to reduce the overhead if needed, which we leave as future work.

---

> ### Comment · Reviewer_5gmS · 2025-08-09
> **Further response**
>
> Thank you for the response. I think method’s substantially higher latency compared to other approaches may limit its applicability in low-latency settings. While further PCA optimizations are suggested, they are left as future work. I will update my score. But given these practical constraints, I update the score from 3 to 4.

---

> > ### Author Response · Authors · 2025-08-09
> >
> > Thank you for recommending our paper and for your thoughtful feedback. We’re encouraged by your updated assessment and will include discussion on the points you raised in the revision.

---

### Official Review · Reviewer_ptsx · 2025-07-05

**Clarity:** 3
**Significance:** 3
**Originality:** 3
**Rating:** 4
**Confidence:** 3

**Summary:**

This paper actively reduces non-causal features in TTA. To identify non-causal features, the authors analyzed how these representations change when applying targeted perturbations to the input data.

**Questions:**

How to handle the limitations mentioned in the above weakness part?

**Ethical Concerns:**

["NO or VERY MINOR ethics concerns only"]

**Final Justification:**

I appreciate the authors' detailed response. My concern has been addressed.  I will raise the score.

**Limitations:**

yes

**Paper Formatting Concerns:**

none.

**Quality:**

3

**Strengths And Weaknesses:**

* Strenghts:

1. The paper proposes a novel causal trimming approach to remove non-causal features from representations, improving model robustness under distribution shifts.

2. This idea is novel in TTA research field. The theoretical analysis is solid.

* Weakness:

1. Designing data augmentations that preserve causal features while perturbing non-causal ones requires domain knowledge. Consequently, this limits the method’s generalizability and practical deployment in domains where such knowledge is unavailable or hard to encode.

2. The identification of non-causal components is based on PCA, which assumes linearity and orthogonality of the principal directions. This may fail in practice.

---

> ### Author Rebuttal · Authors · 2025-07-31
>
> Thank you for the thoughtful feedback and for recognizing our novelty and theoretical analysis.
> The limitations raised in the review were acknowledged in section 7 of the main paper. We discuss the two limitations in more detail below.
>
> **1. Domain knowledge to design augmentation**
>
> We agree that requiring prior knowledge of the data is a limitation of the proposed method.
> Under the setting of TTA, where only a trained model and test data are given, in the absence of any task-specific or domain knowledge, it is inherently difficult—if not impossible—to distinguish causal from non-causal features [1, 2].
>
> Data augmentation is one approach to identifying non-causal features. In practice, when insufficient domain knowledge is given and the optimal augmentation strategies are unknown, the augmentations can be treated as a hyperparameter to search over.
> Specifically, our basic assumption is that, given a dataset and a task,  we at least have the knowledge about the prediction target. Based on this, we can design augmentations that are semantically invariant with respect to the prediction target.
> For example, for the commonly studied image classification task, we recommend searching over general image augmentations, such as AutoAugment and RandomAugment. These augmentations preserve the key causal features, particularly the shape information of objects [3].  Our experiments in the main paper of using them for ImageNet-R and ImageNet-V2 show state-of-the-art performance.
> Birdcall's audio clip is converted to Mel spectrogram, which is treated as image as well.
> Our experiment of using AutoAugment and RandomAugment with Birdcalls and Camelyon show that they can also offer performance gain compared to no adaptation.
> | Augmentation                    | Birdcalls      | Camelyon17     |
> |---------------------------------|----------------|----------------|
> | no TTA                          | 22.74          | 62.31          |
> | Stain color jitter/color jitter | 29.99$\pm$0.80 | 70.17$\pm$0.05 |
> | AutoAugment                     | 27.45$\pm$1.28 | 66.42$\pm$0.01 |
> | RandomAugment                   | 31.76$\pm$0.19 | 65.60$\pm$0.08 |
>
> The paper wants to highlight that reducing non-causal features by adjusting the learned representations is helpful for TTA.  Alternative to data augmentation, if other information, such as a small set of labeled data from multiple distributions, is given, non-causal features can also be identified by comparing the representations of data with the same labels.
>
> **2. Linearity and orthogonality of the principal directions**
>
> We use a simplified representation space that satisfies the linearity and orthogonality assumptions of causal and non-causal features to approximate the learned representation space, so that a practical method could be developed.
> Nonetheless, the theoretical analysis in section 5 does not rely on the assumption.
> We show that even when causal and non-causal features are not orthogonal, and causal features are partially trimmed,
> TACT performs well as long as top PCs alone cause the prediction error, or sufficient causal information is preserved after trimming, and the learned decision boundary is directionally aligned with the causal decision boundary.
> Empirically, the models used for TTA in section 6 do not have their representation space explicitly enforced to satisfy the assumption.
>
> [1] Wang et al., Sound and Complete Causal Identification with Latent Variables Given Local Background Knowledge, NeurIPS 2022.
>
> [2] Schölkopf  et al., Toward causal representation learning, Proceedings of the IEEE 2021.
>
> [3] Geirhos et al., ImageNet-trained CNNs are biased towards texture; increasing shape bias improves accuracy and robustness, ICLR 2019.

---

> ### Author Response · Authors · 2025-08-06
>
> Dear Reviewer ptsx,
>
> We sincerely appreciate the feedback you provided.
> We hope that our discussion on the limitations of TACT addresses your concerns.
> We're happy to clarify anything further, so please don't hesitate to share any remaining questions or suggestions you may have.

---

> > ### Author Response · Authors · 2025-08-08
> >
> > Dear Reviewer ptsx,
> >
> > Thank you for the valuable feedback you've provided so far. We understand you may be busy, but since the discussion phase will conclude in less than 24 hours, we would greatly appreciate it if you could share any additional questions or concerns you might have. This would allow us to respond in a timely and constructive manner. Thank you in advance!

---

### Author Response · Authors · 2025-08-09
**Rebuttal Summary**

We are thankful for the time and effort all reviewers dedicated to providing detailed and constructive feedback.
The reviews and discussions are summarized here:

- **Reviewer ptsx** acknowledges the novelty of our method and the solidity of our theoretical analysis. Concerns were raised about the generalizability of the proposed method due to
     - *Domain knowledge and augmentation design*: we discuss the inherent difficulty of identifying non-causal features without domain knowledge and show that augmentations that maintain key causal features can be helpful.

     - *linearity and orthogonality assumption*: we emphasize that TACT is robust theoretically and empirically when the assumption are not exactly satisfied.

- **Reviewer 5gmS** recognizes that our method addresses the root cause of distribution shift backed by a theoretical guarantee, and raises questions for
     - *Augmentation sensitivity and selection strategy*: we include an empirical study to show that general image augmentation is helpful, and suggest treating augmentation as a hyperparameter to search over.

     - *Orthogonality assumption and co-variation of causal and spurious features*: we stress that TACT does not rely on perfect orthogonality or disentanglement theoretically and empirically. We apply augmentation that minimizes the co-variation.

     - *Statistical significance, scalability and latency*: we show that the improvement of our method is overall significant. Our method generally scales well. The increase in latency is a trade-off for the remarkable performance gain.

     - *Number of PCs to project, hyperparameter and performance*: we refer to the paper and clarify the questions.

- **Reviewer QQc2** applauds the proposed method as well as our empirical and theoretical analysis. The discussion is about
     - *Generalizability over assumptions about augmentation and representation space*: we provide additional empirical studies to show that the proposed method is robust to augmentations that maintain key causal features and representation space that is more prone to entanglement. We clarify that TACT does not rely on perfect disentanglement theoretically. We apply augmentation that minimizes the co-variation.

     - *Alternative nonlinear method*: we discuss the alternative methods. We experiment with ICA and show that it is inferior to the current choice, PCA.

     - *Selection of the number of trimmed PCs, and failure case*: we explain the details in the rebuttal.


- **Reviewer DW91** recognizes the solidity of our empirical analysis. Concerns were raised over
     - *Performance of TACT compared to training time augmentation*: we empirically show that training-time augmentation is not always better than performing TACT after training. Moreover, we show that TACT can improve over training time augmentation.

     - *Performance selected by validation data*: we show that TACT overall performs better than SOTA methods when using validation data selection.

     - *Computation cost*: we show that TACT incurs more computational cost, which justifies the trade-off for significant performance improvement.

     - *Remove components based on a threshold*: our experiment shows that it does not improve the performance.

     - *Careful selection of augmentation*: we empirically show that TACT could still be robust when the augmentations used can vary the non-causal features and keep the causal features approximately invariant. General image augmentations generally work well with TACT.

---

### Note · Authors · 2025-08-12

We want to thank all reviewers and the AC for their helpful feedback and constructive engagement. Below, we present our conclusive remarks:

**Contribution**

Reviewers acknowledge that we introduce a novel TTA method, TACT, to address the root cause of distribution shift by reducing non-causal features in representations.
Our extended empirical evaluation and rigorous theoretical analysis are recognized by reviewers, and they jointly established the effectiveness and soundness of TACT.

**Key points addressed during rebuttal**

*Augmentation design and selection*

We propose to use data augmentation that varies non-causal features to identify non-causal features.
For images, instead of overly relying on prior knowledge, TACT generally works well with common image augmentations such as AutoAugment and RandomAugment, as demonstrated by our BirdCall and Camelyon experiments in reply 1 to reviewer ptsx, and ImageNet-R and ImageNet-V2 experiments in the paper.
TACT can still be robust with augmentations that keep causal features approximately invariant.

*Assumption of linearity, orthogonality, and disentanglement of causal and non-causal features in the representation space*

Empirically, the models we used for TTA are not explicitly constrained to satisfy the assumption. PCA’s approximate separation of non-causal features yields consistent performance gains and robustness across diverse datasets. We also demonstrate in reply Q2.1 to reviewer QQc2 that TACT still performs well on larger models when representation entanglement is common.
Our theoretical analysis establishes the conditions for TACT’s effectiveness, independent of the assumption.

*Synergizing with training-time augmentation*

TACT tackles distribution shift at inference, complementing training-time strategies. Its synergy with training-time augmentation, as stated in reply W1 to reviewer DW91, further highlights its effectiveness.

**Conclusion**

Overall, our findings highlight that reducing non-causal features through representation adjustment is an effective TTA strategy. We demonstrate that TACT remains effective with general augmentations and when the representation space only approximately satisfies the assumption, offering a robust and broadly applicable approach that we believe will inspire future work in TTA.

---

### Decision · Program_Chairs · 2025-09-17

**Decision:**

Accept (poster)

**Comment:**

This paper proposes a backpropagation-free test-time adaptation method that mitigates distribution shift. Theoretical analysis provides conditions for error correction and causal preservation, while experiments across five datasets (image, text, audio) demonstrate consistent gains over state-of-the-art TTA methods.

The work introduces a principled and broadly applicable approach to TTA, bridging causality and representation learning. Despite practical challenges, the method’s novelty, theoretical rigor, and strong empirical validation justify acceptance.

During the rebuttal. authors addressed concerns by showing robustness to approximate assumptions, generalization of augmentations, and complementarity with training-time augmentation. Ablations with ICA and additional large-model experiments strengthen claims. Reviewers found responses satisfactory; two raised scores and one maintained a borderline accept.